# LEARNING TRANSFERABLE MOTOR SKILLS WITH HIERARCHICAL LATENT MIXTURE POLICIES

**Dushyant Rao,**[*] **Fereshteh Sadeghi, Leonard Hasenclever, Markus Wulfmeier,**
**Martina Zambelli, Giulia Vezzani, Dhruva Tirumala, Yusuf Aytar, Josh Merel,**[†]
**Nicolas Heess, & Raia Hadsell**
DeepMind, London, UK

## ABSTRACT

For robots operating in the real world, it is desirable to learn reusable behaviours that can effectively be transferred and adapted to numerous tasks and scenarios. We propose an approach to learn abstract motor skills from data using a hierarchical mixture latent variable model. In contrast to existing work, our method exploits a three-level hierarchy of *both* discrete and continuous latent variables, to capture a set of high-level behaviours while allowing for variance in how they are executed. We demonstrate in manipulation domains that the method can effectively cluster offline data into distinct, executable behaviours, while retaining the flexibility of a continuous latent variable model. The resulting skills can be transferred and fine-tuned on new tasks, unseen objects, and from state to vision-based policies, yielding better sample efficiency and asymptotic performance compared to existing skill- and imitation-based methods. We further analyse how and when the skills are most beneficial: they encourage directed exploration to cover large regions of the state space relevant to the task, making them most effective in challenging sparse-reward settings.

## 1 INTRODUCTION

Reinforcement learning is a powerful and flexible paradigm to train embodied agents, but relies on large amounts of agent experience, computation, and time, on each individual task. Learning each task from scratch is inefficient: it is desirable to learn a set of *skills* that can efficiently be reused and adapted to related downstream tasks. This is particularly pertinent for real-world robots, where interaction is expensive and data-efficiency is crucial. There are numerous existing approaches to learn transferable embodied skills, usually formulated as a two-level hierarchy with a high-level controller and low-level skills. These methods predominantly represent skills as being either continuous, such as goal-conditioned (Lynch et al., 2019; Pertsch et al., 2020b) or latent space policies (Haarnoja et al., 2018; Merel et al., 2019; Singh et al., 2021); or discrete, such as mixture or option-based methods (Sutton et al., 1999; Daniel et al., 2012; Florensa et al., 2017; Wulfmeier et al., 2021). Our goal is to combine these perspectives to leverage their complementary advantages.

We propose an approach to learn a three-level skill hierarchy from an offline dataset, capturing both discrete and continuous variations at multiple levels of behavioural abstraction. The model comprises a low-level latent-conditioned controller that can learn motor primitives, a set of continuous latent mid-level skills, and a discrete high-level controller that can compose and select among these abstract mid-level behaviours. Since the mid- and high-level form a mixture, we call our method Hierarchical Latent Mixtures of Skills (HeLMS). We demonstrate on challenging object manipulation tasks that our method can decompose a dataset into distinct, intuitive, and reusable behaviours. We show that these skills lead to improved sample efficiency and performance in numerous transfer scenarios: reusing skills for new tasks, generalising across unseen objects, and transferring from state to vision-based policies. Further analysis and ablations reveal that both continuous and discrete components are beneficial, and that the learned hierarchical skills are most useful in sparse-reward settings, as they encourage directed exploration of task-relevant parts of the state space.

---

[*]Corresponding author. Email: `dushyantr@deepmind.com`
[†]Work done while at DeepMind

Our main contributions are as follows:

- We propose a novel approach to learn skills at different levels of abstraction from an offline dataset. The method captures both discrete behavioural modes and continuous variation using a hierarchical mixture latent variable model.
- We present two techniques to reuse and adapt the learned skill hierarchy via reinforcement learning in downstream tasks, and perform extensive evaluation and benchmarking in different transfer settings: to new tasks and objects, and from state to vision-based policies.
- We present a detailed analysis to interpret the learned skills, understand when they are most beneficial, and evaluate the utility of both continuous and discrete skill representations.

## 2 RELATED WORK

A long-standing challenge in reinforcement learning is the ability to learn reusable motor skills that can be transferred efficiently to related settings. One way to learn such skills is via multi-task reinforcement learning (Heess et al., 2016; James et al., 2018; Hausman et al., 2018; Riedmiller et al., 2018), with the intuition that behaviors useful for a given task should aid the learning of related tasks. However, this often requires careful curation of the task set, where each skill represents a separate task. Some approaches avoid this by learning skills in an unsupervised manner using intrinsic objectives that often maximize the entropy of visited states while keeping skills distinguishable (Gregor et al., 2017; Eysenbach et al., 2019; Sharma et al., 2019; Zhang et al., 2020).

A large body of work explores skills from the perspective of unsupervised segmentation of repeatable behaviours in temporal data (Niekum & Barto, 2011; Ranchod et al., 2015; Krüger et al., 2016; Lioutikov et al., 2017; Shiarlis et al., 2018; Kipf et al., 2019; Tanneberg et al., 2021). Other works investigate movement or motor primitives that can be selected or sequenced together to solve complex manipulation or locomotion tasks (Mülling et al., 2013; Rueckert et al., 2015; Lioutikov et al., 2015; Paraschos et al., 2018; Merel et al., 2020; Tosatto et al., 2021; Dalal et al., 2021). Some of these methods also employ mixture models to jointly model low-level motion primitives and a high-level primitive controller (Muelling et al., 2010; Colomé & Torras, 2018; Pervez & Lee, 2018); the high-level controller can also be implicit and decentralised over the low-level primitives (Goyal et al., 2019).

Several existing approaches employ architectures in which the policy is comprised of two (or more) levels of hierarchy. Typically, a low-level controller represents the learned set of skills, and a high-level policy instructs the low-level controller via a latent variable or goal. Such latent variables can be discrete (Florensa et al., 2017; Wulfmeier et al., 2020) or continuous (Nachum et al., 2018; Haarnoja et al., 2018) and regularization of the latent space is often crucial (Tirumala et al., 2019). The latent variable can represent the behaviour for one timestep, for a fixed number of timesteps (Ajay et al., 2021), or *options* with different durations (Sutton et al., 1999; Bacon et al., 2017; Wulfmeier et al., 2021). One such approach that is particularly relevant (Florensa et al., 2017) learns a diverse set of skills, via a discrete latent variable that interacts multiplicatively with the state to enable continuous variation in a Stochastic Neural Network policy; this skill space is then transferred to locomotion tasks by learning a new categorical controller. Our method differs in a few key aspects: our proposed three-level hierarchical architecture explicitly models abstract discrete skills while allowing for temporal dependence and lower-level latent variation in their execution, enabling diverse object-centric behaviours in challenging manipulation tasks.

Our work is related to methods that learn robot policies from demonstrations (LfD, e.g. (Rajeswaran et al., 2018; Shiarlis et al., 2018; Strudel et al., 2020)) or more broadly from logged data (offline RL, e.g. (Wu et al., 2019; Kumar et al., 2020; Wang et al., 2020)). While many of these focus on learning single-task policies, several approaches learn skills offline that can be transferred online to new tasks (Merel et al., 2019; Lynch et al., 2019; Pertsch et al., 2020a; Ajay et al., 2021; Singh et al., 2021). These all train a two-level hierarchical model, with a high-level encoder that maps to a continuous latent space, and a low-level latent-conditioned controller. The high-level encoder can encode a whole trajectory (Pertsch et al., 2020a; 2021; Ajay et al., 2021); a short look-ahead state sequence (Merel et al., 2019); the current and final goal state (Lynch et al., 2019); or can even be simple isotropic Gaussian noise (Singh et al., 2021) that can be flexibly transformed by a flow-based low-level controller. At transfer time, a new high-level policy is learned from scratch: this can be more efficient with skill priors (Pertsch et al., 2020a) or temporal abstraction (Ajay et al., 2021).

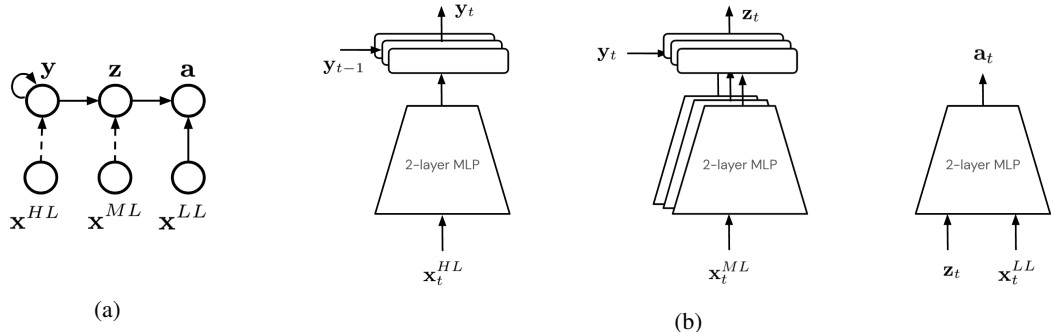

(a)  (b)

Figure 1: (a) Graphical model for HeLMS, with solid lines indicating the underlying generative model (prior) and dashed lines indicating dependencies introduced by the inference model (posterior). (b) Network architecture, showing the high-, mid-, and low-level networks from left to right, respectively. As indicated by superscripts, different subsets of the input state $\mathbf{x}$ can be provided to the high level ($HL$), mid level ($ML$), and low level ($LL$) (information-asymmetry).

HeLMS builds on this large body of work by explicitly modelling both discrete and continuous behavioural structure via a three-level skill hierarchy. We use similar information asymmetry to Neural Probabilistic Motor Primitives (NPMP) (Merel et al., 2019; 2020), conditioning the high-level encoder on a short look-ahead trajectory. However HeLMS explicitly captures discrete modes of behaviour via the high-level controller, and learns an additional mid-level which is able to transfer abstract skills to downstream tasks, rather than learning a continuous latent policy from scratch.

## 3  METHOD

This paper examines a two-stage problem setup: an offline stage where a hierarchical skill space is learned from a dataset, and an online stage where these skills are transferred to a reinforcement learning setting. The dataset $\mathcal{D}$ comprises a set of trajectories, each a sequence of state-action pairs $\{\mathbf{x}_t, \mathbf{a}_t\}_{t=0}^{T}$. The model incorporates a discrete latent variable $\mathbf{y}_t \in \{1, \ldots, K\}$ as a high-level *skill selector* (for a fixed number of skills $K$), and a mid-level continuous variable $\mathbf{z}_t \in \mathbb{R}^{n_z}$ conditioned on $\mathbf{y}_t$ which *parameterises* each skill. Marginally, $\mathbf{z}_t$ is then a latent mixture distribution representing both a discrete set of skills and the variation in their execution. A sample of $\mathbf{z}_t$ represents an abstract behaviour, which is then executed by a low-level controller $p(\mathbf{a}_t \mid \mathbf{z}_t, \mathbf{x}_t)$. The learned skill space can then be transferred to a reinforcement learning agent $\pi$ in a Markov Decision Process defined by tuple $\{\mathcal{S}, \mathcal{A}, \mathcal{T}, \mathcal{R}, \gamma\}$: these represent the state, action, and transition distributions, reward function, and discount factor respectively. When transferring, we train a new high-level controller that acts either at the level of discrete skills $\mathbf{y}_t$ or continuous $\mathbf{z}_t$, and freeze lower levels of the policy.

We explain our method in detail in the following sections.

### 3.1  LATENT MIXTURE SKILL SPACES FROM OFFLINE DATA

Our method employs the generative model in Figure 1a. As shown, the state inputs can be different for each level of the hierarchy, but to keep notation uncluttered, we refer to all state inputs as $\mathbf{x}_t$ and the specific input can be inferred from context. The joint distribution of actions and latents over a trajectory is decomposed into a latent prior $p(\mathbf{y}_{0:T}, \mathbf{z}_{1:T})$ and a low-level controller $p(\mathbf{a}_t \mid \mathbf{z}_t, \mathbf{x}_t)$:

$$p(\mathbf{a}_{1:T}, \mathbf{y}_{0:T}, \mathbf{z}_{1:T} \mid \mathbf{x}_{1:T}) = p(\mathbf{y}_{0:T}, \mathbf{z}_{1:T}) \prod_{t=1}^{T} p(\mathbf{a}_t \mid \mathbf{z}_t, \mathbf{x}_t)$$

$$p(\mathbf{y}_{0:T}, \mathbf{z}_{1:T}) = p(\mathbf{y}_0) \prod_{t=1}^{T} p(\mathbf{y}_t \mid \mathbf{y}_{t-1}) p(\mathbf{z}_t \mid \mathbf{y}_t). \tag{1}$$

Intuitively, the categorical variable $\mathbf{y}_t$ can capture discrete modes of behaviour, and the continuous latent $\mathbf{z}_t$ is conditioned on this to vary the execution of each behaviour. Thus, $\mathbf{z}_t$ follows a mixture

distribution, encoding all the relevant information on desired abstract behaviour for the low-level controller $p(\mathbf{a}_t \mid \mathbf{z}_t, \mathbf{x}_t)$. Since each categorical latent $\mathbf{y}_t$ is dependent on $\mathbf{y}_{t-1}$, and $\mathbf{z}_t$ is only dependent on $\mathbf{y}_t$, this prior can be thought of as a Hidden Markov model over the sequence of $\mathbf{z}_{1:T}$.

To perform inference over the latent variables, we introduce the variational approximation:

$$q(\mathbf{y}_{0:T}, \mathbf{z}_{1:T} \mid \mathbf{x}_{1:T}) = p(\mathbf{y}_0) \prod_{t=1}^{T} q(\mathbf{y}_t \mid \mathbf{y}_{t-1}, \mathbf{x}_t) q(\mathbf{z}_t \mid \mathbf{y}_t, \mathbf{x}_t) \tag{2}$$

Here, the selection of a skill $\mathbf{y}_t \sim q(\mathbf{y}_t \mid \mathbf{y}_{t-1}, \mathbf{x}_t)$ is dependent on that of the previous timestep (allowing for temporal consistency), as well as the input. The mid-level skill is then parameterised by $\mathbf{z}_t \sim q(\mathbf{z}_t \mid \mathbf{y}_t, \mathbf{x}_t)$ based on the chosen skill and current input. $p(\mathbf{y}_0)$ and $p(\mathbf{y}_t \mid \mathbf{y}_{t-1})$ model a skill prior and skill transition prior respectively, while $p(\mathbf{z}_t \mid \mathbf{y}_t)$ represents a skill parameterisation prior to regularise each mid-level skill. While all of these priors can be learned in practice, we only found it necessary to learn the transition prior, with a uniform categorical for the initial skill prior and a simple fixed $\mathcal{N}(0, I)$ prior for $p(\mathbf{z}_t \mid \mathbf{y}_t)$.

**Training via the Evidence Lower Bound**   The proposed model contains a number of components with trainable parameters: the prior parameters $\psi = \{\psi_a, \psi_y\}$ for the low-level controller and categorical transition prior respectively; and posterior parameters $\phi = \{\phi_y, \phi_z\}$ for the high-level controller and mid-level skills. For a trajectory $\{\mathbf{x}_{1:T}, \mathbf{a}_{1:T}\} \sim \mathcal{D}$, we can compute the Evidence Lower Bound for the state-conditional action distribution, $ELBO \leq p(\mathbf{a}_{1:T} \mid \mathbf{x}_{1:T})$, as follows:

$$ELBO = \mathbb{E}_{q_\phi(\mathbf{y}_{0:T}, \mathbf{z}_{1:T} \mid \mathbf{x}_{1:T})} \Big[ \log p_\psi(\mathbf{a}_{1:T}, \mathbf{y}_{0:T}, \mathbf{z}_{1:T} \mid \mathbf{x}_{1:T}) - \log q_\phi(\mathbf{y}_{0:T}, \mathbf{z}_{1:T} \mid \mathbf{x}_{1:T}) \Big]$$

$$\approx \sum_{t=1}^{T} \Bigg[ \sum_{\mathbf{y}_t} q(\mathbf{y}_t \mid \mathbf{x}_{1:t}) \Big( \overbrace{\log p_{\psi_a}(\mathbf{a}_t \mid \tilde{\mathbf{z}}_t^{\{\mathbf{y}_t\}}, \mathbf{x}_t)}^{\text{per-component action recon}} - \beta_z \overbrace{\mathrm{KL}(q_{\phi_z}(\mathbf{z}_t \mid \mathbf{y}_t, \mathbf{x}_t) \,\|\, p(\mathbf{z}_t \mid \mathbf{y}_t))}^{\text{per-component KL regulariser}} \Big) \Bigg]$$

$$-\beta_y \sum_{t=1}^{T} \Bigg[ \sum_{\mathbf{y}_{t-1}} q(\mathbf{y}_{t-1} \mid \mathbf{x}_{1:t-1}) \underbrace{\mathrm{KL}\big(q_{\phi_y}(\mathbf{y}_t \mid \mathbf{y}_{t-1}, \mathbf{x}_t) \,\|\, p_{\psi_y}(\mathbf{y}_t \mid \mathbf{y}_{t-1})\big)}_{\text{categorical regulariser}} \Bigg] \tag{3}$$

where $\tilde{\mathbf{z}}_t^{\{\mathbf{y}_t\}} \sim q(\mathbf{z}_t \mid \mathbf{y}_t, \mathbf{x}_t)$. The coefficients $\beta_y$ and $\beta_z$ can be used to weight the KL terms, and the cumulative component probability $q(\mathbf{y}_t \mid \mathbf{x}_{1:t})$ can be computed iteratively as $q(\mathbf{y}_t \mid \mathbf{x}_{1:t}) = \sum_{\mathbf{y}_{t-1}} q_{\phi_y}(\mathbf{y}_t \mid \mathbf{y}_{t-1}, \mathbf{x}_t) q(\mathbf{y}_{t-1} \mid \mathbf{x}_{1:t-1})$. In other words, for each timestep $t$ and each mixture component, we compute the latent sample and the corresponding action log-probability, and the KL-divergence between the component posterior and prior. This is then marginalised over all $\mathbf{y}_t$, with an additional KL over the categorical transitions. For more details, see Appendix C.

**Information-asymmetry**   As noted in previous work (Tirumala et al., 2019; Galashov et al., 2019), hierarchical approaches often benefit from information-asymmetry, with higher levels seeing additional context or task-specific information. This ensures that the high-level remains responsible for abstract, task-related behaviours, while the low-level executes simpler motor primitives. We employ similar techniques in HeLMS: the low-level inputs $\mathbf{x}_{LL}$ comprise the proprioceptive state of the embodied agent; the mid-level inputs $\mathbf{x}_{ML}$ also include the poses of objects in the environment; and the high-level $\mathbf{x}_{HL}$ concatenates both object and proprioceptive state for a short number of lookahead timesteps. The high- and low-level are similar to (Merel et al., 2019), with the low-level controller enabling motor primitives based on proprioceptive information, and the high-level using the lookahead information to provide additional context regarding behavioural intent when specifying which skill to use. The key difference is the categorical high-level and the additional mid-level, with which HeLMS can learn more object-centric skills and transfer these to downstream tasks.

**Network architectures**   The architecture and information flow in HeLMS are shown in Figure 1b. The high-level network contains a *gated head*, which uses the previous skill $\mathbf{y}_{t-1}$ to index into one of $K$ categorical heads, each of which specify a distribution over $\mathbf{y}_t$. For a given $\mathbf{y}_t$, the corresponding mid-level skill network is selected and used to sample a latent action $\mathbf{z}_t$, which is then used as input for the latent-conditioned low-level controller, which parameterises the action distribution. The skill transition prior $p(\mathbf{y}_t \mid \mathbf{y}_{t-1})$ is also learned, and is parameterised as a linear softmax layer which takes in a one-hot representation of $\mathbf{y}_{t-1}$ and outputs the distribution over $\mathbf{y}_t$. All components are trained end-to-end via the objective in Equation 3.

## 3.2 REINFORCEMENT LEARNING WITH RELOADED SKILLS

Once learned, we propose two methods to transfer the hierarchical skill space to downstream tasks. Following previous work (e.g. (Merel et al., 2019; Singh et al., 2021)), we freeze the low-level controller $p(\mathbf{a}_t \,|\, \mathbf{z}_t, \mathbf{x}_t)$, and learn a policy for either the continuous ($\mathbf{z}_t$) or discrete ($\mathbf{y}_t$) latent.

**Categorical agent**   One simple and effective technique is to additionally freeze the mid-level components $q(\mathbf{z}_t \,|\, \mathbf{y}_t, \mathbf{x}_t)$, and learn a categorical high-level controller $\pi(\mathbf{y}_t \,|\, \mathbf{x}_t)$ for the downstream task. The learning objective is given by:

$$\mathcal{J} = \mathbb{E}_\pi \left[ \sum_t \gamma^t \left( r_t - \eta_y \mathrm{KL}(\pi(\mathbf{y}_t \,|\, \mathbf{x}_t) \,||\, \pi_0(\mathbf{y}_t \,|\, \mathbf{x}_t)) \right) \right], \tag{4}$$

where the standard discounted return objective in RL is augmented by an additional term performing KL-regularisation to some prior $\pi_0$ scaled by coefficient $\eta_y$. This could be any categorical distribution such as the previously learned transition prior $p(\mathbf{y}_t \,|\, \mathbf{y}_{t-1})$, but in this paper we regularise to the uniform categorical prior to encourage diversity. While any RL algorithm could be used to optimize $\pi(\mathbf{y}_t \,|\, \mathbf{x}_t)$, in this paper we use MPO (Abdolmaleki et al., 2018) with a categorical action distribution (see Appendix B for details). We hypothesise that this method improves sample efficiency by converting a continuous control problem into a discrete abstract action space, which may also aid in credit assignment. However, since both the mid-level components and low-level are frozen, it can limit flexibility and plasticity, and also requires that all of the mid- and low-level input states are available in the downstream task. We call this method **HeLMS-cat**.

**Mixture agent**   A more flexible method of transfer is to train a latent mixture policy, $\pi(\mathbf{z}_t \,|\, \mathbf{x}_t) = \sum_{\mathbf{y}_t} \pi(\mathbf{y}_t \,|\, \mathbf{x}_t) \pi(\mathbf{z}_t \,|\, \mathbf{y}_t, \mathbf{x}_t)$. In this case, the learning objective is given by:

$$\mathcal{J} = \mathbb{E}_\pi \left[ \sum_t \gamma^t \left( r_t - \eta_y \mathrm{KL}(\pi(\mathbf{y}_t \,|\, \mathbf{x}_t) \,||\, \pi_0(\mathbf{y}_t \,|\, \mathbf{x}_t)) - \eta_z \sum_{\mathbf{y}_t} \mathrm{KL}(\pi(\mathbf{z}_t \,|\, \mathbf{y}_t, \mathbf{x}_t) \,||\, \pi_0(\mathbf{z}_t \,|\, \mathbf{y}_t, \mathbf{x}_t)) \right) \right], \tag{5}$$

where in addition to the categorical prior, we also regularise each mid-level skill to a corresponding prior $\pi_0(\mathbf{z}_t \,|\, \mathbf{y}_t, \mathbf{x}_t)$. While the priors could be any policies, we set them to be the skill posteriors $q(\mathbf{z}_t \,|\, \mathbf{y}_t, \mathbf{x}_t)$ learned offline, to ensure the mixture components remain close to the pre-learned skills. This is related to (Tirumala et al., 2019), which also applies KL-regularisation at multiple levels of a hierarchy. While the high-level controller $\pi(\mathbf{y}_t \,|\, \mathbf{x}_t)$ is learned from scratch, the mixture components can also be initialised to $q(\mathbf{z}_t \,|\, \mathbf{y}_t, \mathbf{x}_t)$, to allow for initial exploration over the space of skills. Alternatively, the mixture components can use different inputs, such as vision: this setup allows vision-based skills to be learned efficiently by regularising to state-based skills learned offline. We optimise this using RHPO (Wulfmeier et al., 2020), which employs a similar underlying optimisation to MPO for mixture policies (see Appendix B for details). We call this **HeLMS-mix**.

## 4 EXPERIMENTS

Our experiments focus on the following questions: (1) Can we learn a hierarchical latent mixture skill space of distinct, interpretable behaviours? (2) How do we best reuse this skill space to improve sample efficiency and performance on downstream tasks? (3) Can the learned skills transfer effectively to multiple downstream scenarios: (i) different objects; (ii) different tasks; and (iii) different modalities such as vision-based policies? (4) How exactly do these skills aid learning of downstream manipulation tasks? Do they aid exploration? Are they useful in sparse or dense reward scenarios?

### 4.1 EXPERIMENTAL SETUP

**Environment and Tasks**   We focus on manipulation tasks, using a MuJoCo-based environment with a single Sawyer arm, and three objects coloured red, green, and blue. We follow the challenging object stacking benchmark of Lee et al. (2021), which specifies five object sets (Figure 2), carefully designed to have diverse geometries and present different challenges for a stacking agent. These range from simple rectangular objects (object set 4), to geometries such as slanted faces (sets 1 and 2) that make grasping or stacking the objects more challenging. This environment allows us

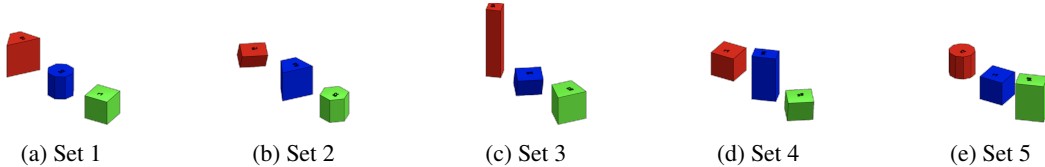

(a) Set 1      (b) Set 2      (c) Set 3      (d) Set 4      (e) Set 5

Figure 2: The object sets (triplets) used for our experiments, introduced by (Lee et al., 2021).

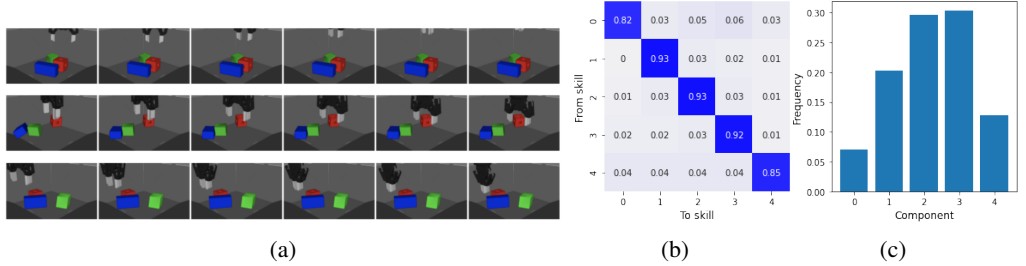

(a)      (b)      (c)

Figure 3: (a) Image sequences showing example rollouts when fixing the discrete skill (different for each row) and running the mid- and low-level controllers in the environment. Each skill executes a different behaviour, such as lifting (top row), reach-to-red (middle row), or grasping (bottom row) (b) The learned skill transition prior $p(\mathbf{y}_t \mid \mathbf{y}_{t-1})$. (c) Histogram showing the use of different skills.

to systematically evaluate generalisation of manipulation behaviours for different tasks interacting with geometrically different objects. For further information, we refer the reader to Appendix D.1 or to (Lee et al., 2021). Details of the rewards for the different tasks are also provided in Appendix F.

**Datasets**     To evaluate our approach and baselines in the manipulation settings, we use two datasets:

- `red_on_blue_stacking`: this data is collected by an agent trained to stack the red object on the blue object and ignore the green one, for the simplest object set, set4.
- `all_pairs_stacking`: similar to the previous case, but with all six pairwise stacking combinations of {red, green, blue}, and covering all of the five object sets.

**Baselines**     For evaluation in transfer scenarios, we compare HeLMS with a number of baselines:

- **From scratch**: We learn the task from scratch with MPO, without an offline learning phase.
- **NPMP+KL**: We compare against NPMP (Merel et al., 2019), which is the most similar skill-based approach in terms of information-asymmetry and policy conditioning. We make some small changes to the originally proposed method, and also apply additional KL-regularisation to the latent prior: we found this to improve performance significantly in our experiments. For more details and an ablation, see Appendix A.2.
- **Behaviour Cloning (BC)**: We apply behaviour cloning to the dataset, and fine-tune this policy via MPO on the downstream task. While the actor is initialised to the solution obtained via BC, the critic still needs to be learned from scratch.
- **Hierarchical BC**: We evaluate a hierarchical variant of BC with a similar latent space **z** to NPMP using a latent Gaussian high-level controller. However, rather than freezing the low-level and learning just a high-level policy, Hierarchical BC fine-tunes the entire model.
- **Asymmetric actor-critic**: For state-to-vision transfer, HeLMS uses prior skills that depend on object states to learn a purely vision-based policy. Thus, we also compare against a variant of MPO with an asymmetric actor-critic (Pinto et al., 2017) setup, which uses object states differently: to speed up learning of the critic, while still learning a vision-based actor.

## 4.2   LEARNING SKILLS FROM OFFLINE DATA

We first aim to understand whether we can learn a set of distinct and interpretable skills from data (question (1)). For this, we train HeLMS on the `red_on_blue_stacking` dataset with 5 skills.

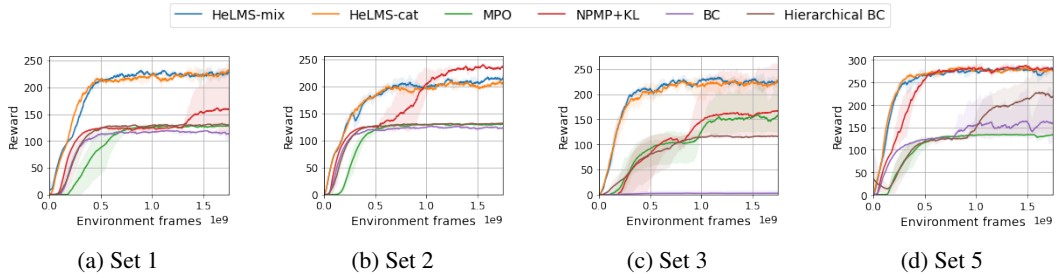

Figure 4: Performance when transferring to the red-on-blue stacking task using staged sparse reward with every unseen object set.

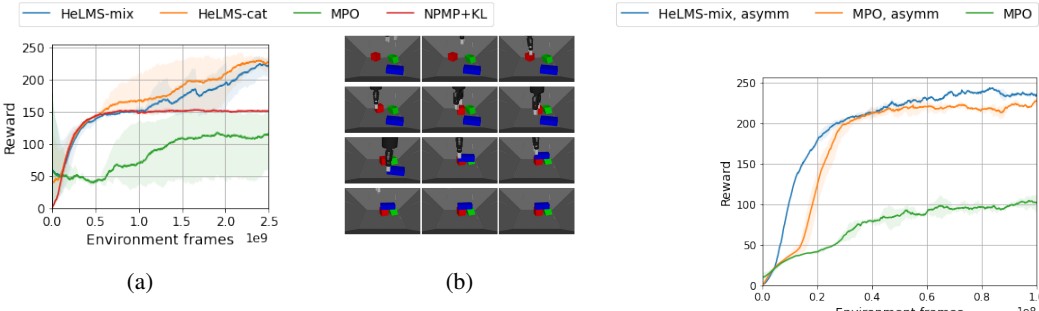

Figure 5: (a) Performance on pyramid task; and (b) image sequence showing episode rollout from a learned solution on this task (left-to-right, top-to-bottom).

Figure 6: Performance for vision-based stacking.

Figure 3a shows some example episode rollouts when the learned hierarchical agent is executed in the environment, holding the high-level categorical skill constant for an episode. Each row represents a different skill component, and the resulting behaviours are both distinct and diverse: for example, a lifting skill (row 1) where the gripper closes and rises up, a reaching skill (row 2) where the gripper moves to the red object, or a grasping skill (row 3) where the gripper lowers and closes its fingers. Furthermore, without explicitly encouraging this, the emergent skills capture temporal consistency: Figure 3b shows the learned prior $p(\mathbf{y}_t \,|\, \mathbf{y}_{t-1})$ (visualised as a transition matrix) assigns high probability along the diagonal (remaining in the same skill). Finally, Figure 3c demonstrates that all skills are used, without degeneracy.

## 4.3 TRANSFER TO DOWNSTREAM TASKS

**Generalising to different objects** We next evaluate whether the previously learned skills (i.e. trained on the simple objects in set 4) can effectively transfer to more challenging object interaction scenarios: the other four object sets proposed by (Lee et al., 2021). The task uses a sparse staged reward, with reward incrementally given after completing each sub-goal of the stacking task. As shown in Figure 4, both variants of HeLMS learn significantly faster than baselines on the different object sets. Compared to the strongest baseline (NPMP), HeLMS reaches better average asymptotic performance (and much lower variance) on two object sets (1 and 3), performs similarly on set 5, and does poorer on object set 2. The performance on object set 2 potentially highlights a trade-off between incorporating higher-level abstract behaviours and maintaining low-level flexibility: this object set often requires a reorientation of the bottom object due to its slanted faces, a behaviour that is not common in the offline dataset, which might require greater adaptation of mid- and low-level skills. This is an interesting investigation we leave for future work.

**Compositional reuse of skills** To evaluate whether the learned skills are composable for new tasks, we train HeLMS on the `all_pairs_stacking` dataset with 10 skills, and transfer to a *pyramid* task. In this setting, the agent has to place the red object adjacent to the green object, and stack the blue object on top to construct a pyramid. The task is specified via a sparse staged reward

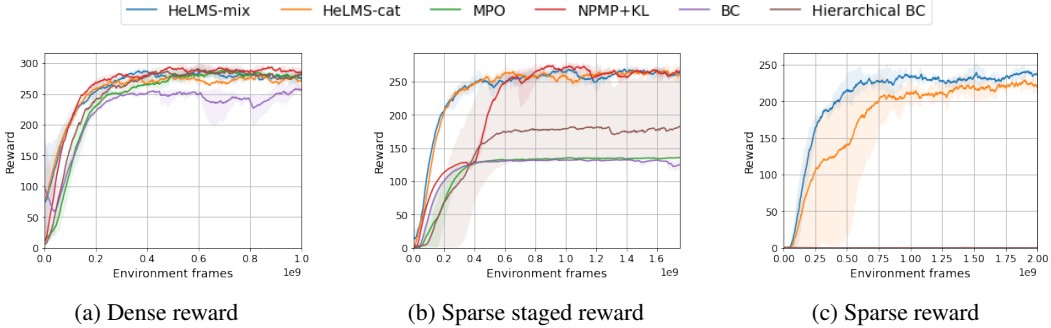

(a) Dense reward  (b) Sparse staged reward  (c) Sparse reward

Figure 7: Transfer performance on a stacking task with different reward sparsities. Left: dense reward, Centre: staged sparse reward, Right: sparse reward

for each stage or sub-task: reaching, grasping, lifting, and placing the red object, and subsequently the blue object. In Figure 5(a), we plot the performance of both variants of our approach, as well as NPMP and MPO; we omit the BC baselines as this involves transferring to a fundamentally different task. Both HeLMS-mix and HeLMS-cat reach a higher asymptotic performance than both NPMP and MPO, indicating that the learned skills can be better transferred to a different task. We show an episode rollout in Figure 5(b) in which the learned agent can successfully solve the task.

**From state to vision-based policies**   While our method learns skills from proprioception and object state, we evaluate whether these skills can be used to more efficiently learn a vision-based policy. This is invaluable for practical real-world scenarios, since the agent acts from pure visual observation at test time without requiring privileged and often difficult-to-obtain object state information.

We use the HeLMS-mix variant to transfer skills to a vision-based policy, by reusing the low-level controller, initialising a new high-level controller and mid-level latent skills (with vision and proprioception as input), and KL-regularising these to the previously learned state-based skills. While the learned policy is vision-based, this KL-regularisation still assumes access to object states during training. For a fair comparison, we additionally compare our approach with a version of MPO using an asymmetric critic (Pinto et al., 2017), which exploits object state information instead of vision in the critic, and also use this for HeLMS. As shown in Figure 6, learning a vision-based policy with MPO from scratch is very slow and computationally intensive, but an asymmetric critic significantly speeds up learning, supporting the empirical findings of Pinto et al. (2017). However, HeLMS once again demonstrates better sample efficiency, and reaches slightly better asymptotic performance. We note that this uses the same offline model as for the object generalisation experiments, showing that the same state-based skill space can be reused in numerous settings, even for vision-based tasks.

## 4.4   WHERE AND HOW CAN HIERARCHICAL SKILL REUSE BE EFFECTIVE?

**Sparse reward tasks**   We first investigate how HeLMS performs for different rewards: a dense shaped reward, the sparse staged reward from the object generalisation experiments, and a fully sparse reward that is only provided after the agent stacks the object. For this experiment, we use the skill space trained on `red_on_blue_stacking` and transfer it to the same RL task of stacking on object set 4. The results are shown in Figure 7. With a dense reward (and no object transfer required), all of the approaches can successfully learn the task. With the sparse staged reward, the baselines all plateau at a lower performance, with the exception of NPMP, as previously discussed. However, for the challenging fully-sparse scenario, HeLMS is the only method that achieves non-zero reward. This neatly illustrates the benefit of the proposed hierarchy of skills: it allows for directed exploration which ensures that even sparse rewards can be encountered. This is consistent with observations from prior work in hierarchical reinforcement learning (Florensa et al., 2017; Nachum et al., 2019), and we next investigate this claim in more depth for our manipulation setting.

**Exploration**   To measure whether the proposed approach leads to more directed exploration, we record the average coverage in state space at the start of RL (i.e. zero-shot transfer). This is computed as the variance (over an episode) of the state $\mathbf{x}_t$, separated into three interpretable groups:

| Method | Reward | | State coverage ($\times 10^{-2}$) | | |
|---|---|---|---|---|---|
| | Dense | Staged | Joints | Grasp | Objects |
| MPO | 3.16 | 0.0 | 8.72 | 0.004 | 1.21 |
| NPMP | 3.67 | 0.0 | 3.43 | 0.0 | 1.45 |
| BC | 31.68 | 0.004 | 4.22 | 0.05 | 1.52 |
| Hier. BC | 16.42 | 0.004 | 2.61 | 0.04 | 1.31 |
| HeLMS | 20.46 | 0.05 | 2.98 | 1.10 | 1.61 |

Table 1: Analysis of zero-shot exploration at the start of RL, in terms of reward and state coverage (variance over an episode of different subsets of the agent's state). Results are averaged over 1000 episodes.

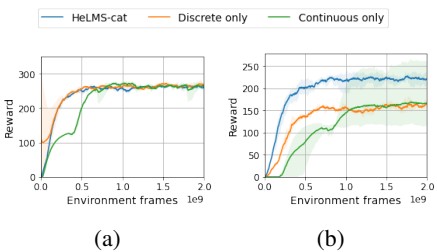

(a)  (b)

Figure 8: Ablation for continuous and discrete components during offline learning, when transferring to the (a) easy case (object set 4) and (b) hard case (object set 3).

joints (angles and velocity), grasp (a simulated grasp sensor), and object poses. We also record the total reward (dense and sparse staged). The results are reported in Table 1. While all approaches achieve some zero-shot dense reward (with BC the most effective), HeLMS[1] receives a sparse staged reward an order of magnitude greater. Further, in this experiment we found it was able to achieve the fully sparse reward (stacked) in one episode. Analysing the state coverage results, while other methods are able to cover the joint space more (e.g. by randomly moving the joints), HeLMS is nearly two orders of magnitude higher for grasp states. This indicates the utility of hierarchical skills: by acting over the space of abstract skills rather than low-level actions, HeLMS performs directed exploration and targets particular states of interest, such as grasping an object.

### 4.5 Ablation Studies

**Capturing continuous and discrete structure**  To evaluate the benefit of both continuous and discrete components, we train our method with a fixed variance of zero for each latent component (i.e. 'discrete-only') and transfer to the stacking task with sparse staged reward in an easy case (object set 4) and hard case (object set 3), as shown in Figure 8(a) and (b). We also evaluate the 'continuous-only' case with just a single Gaussian to represent the high- and mid-level skills: this is equivalent to the NPMP+KL baseline. We observe the the discrete component alone leads to improved sample efficiency in both cases, but modelling both discrete and continuous latent behaviours makes a significant difference in the hard case. In other words, when adapting to challenging objects, it is important to capture discrete skills, but allow for latent variation in how they are executed.

**KL-regularisation**  We also perform an ablation for KL-regularisation during the offline phase (via $\beta_z$) and online RL (via $\eta_z$), to gauge the impact on transfer; see Appendix A.1 for details.

### 5 Conclusion

We present HeLMS, an approach to learn transferable and reusable skills from offline data using a hierarchical mixture latent variable model. We analyse the learned skills to show that they effectively cluster data into distinct, interpretable behaviours. We demonstrate that the learned skills can be flexibly transferred to different tasks, unseen objects, and to different modalities (such as from state to vision). Ablation studies indicate that it is beneficial to model both discrete modes and continuous variation in behaviour, and highlight the importance of KL-regularisation when transferring to RL and fine-tuning the entire mixture of skills. We also perform extensive analysis to understand where and how the proposed skill hierarchy can be most useful: we find that it is particularly invaluable in sparse reward settings due to its ability to perform directed exploration.

There are a number of interesting avenues for future work. While our model demonstrated temporal consistency, it would be useful to more actively encourage and exploit this for sample-efficient transfer. It would also be useful to extend this work to better fine-tune lower level behaviours, to allow for flexibility while exploiting high-level behavioural abstractions.

---

[1] Note that HeLMS-cat and HeLMS-mix are identical for this analysis: at the start of reinforcement learning, both variants transfer the mid-level skills while initialising a new high-level controller.

ACKNOWLEDGMENTS

The authors would like to thank Coline Devin for detailed comments on the paper and for generating the `all_pairs_stacking` dataset. We would also like to thank Alex X. Lee and Konstantinos Bousmalis for help with setting up manipulation experiments. We are also grateful to reviewers for their feedback.

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

# APPENDIX

## A ADDITIONAL EXPERIMENTS

### A.1 ABLATIONS FOR KL-REGULARISATION

In these experiments, we investigate the effect of KL-regularisation on the mid-level components, both for the offline learning phase (regularising each component to $p(\mathbf{z}_t \,|\, \mathbf{y}_t) = \mathcal{N}(0, I)$ via coefficient $\beta_z$), and the online reinforcement learning stage via HeLMS-mix (regularising each component to the mid-level skills learned offline, via coefficient $\eta_z$)). The results are reported in Figure 9, where each plot represents a different setting for offline KL-regularisation (either regularisation to $\mathcal{N}(0, I)$ with $\beta_z = 0.01$, or no regularisation with $\beta_z = 0$) and a different transfer case (the easy case of transferring to object set 4, or the hard case of transferring to object set 3). Each plot shows the downstream performance when varying the strength of KL-regularisation during RL via coefficient $\eta_z$. The HeLMS-cat approach represents the extreme case where the skills are entirely frozen (i.e. full regularisation).

The results suggest some interesting properties of the latent skill space based on regularisation. When regularising the mid-level components to the $\mathcal{N}(0, I)$ prior, it is important to regularise during online RL; this is especially true for the hard transfer case, where HeLMS-cat performs much better, and the performance degrades significantly with lower regularisation values. However, when removing mid-level regularisation during offline learning, the method is insensitive to regularisation during RL over the entire range evaluated, from $0.01$ to $100.0$. We conjecture that with mid-level skills regularised to $\mathcal{N}(0, I)$, the different mid-level skills are drawn closer together and occupy a more compact region in latent space, such that KL-regularisation is necessary during RL for a skill to avoid drifting and overlapping with the latent distribution of other skills (i.e. skill degeneracy). In contrast, without offline KL-regularisation, the skills are free to expand and occupy more distant regions of the latent space, rendering further regularisation unnecessary during RL. Such latent space properties could be further analysed to improve learning and transfer of skills; we leave this as an interesting direction for future work.

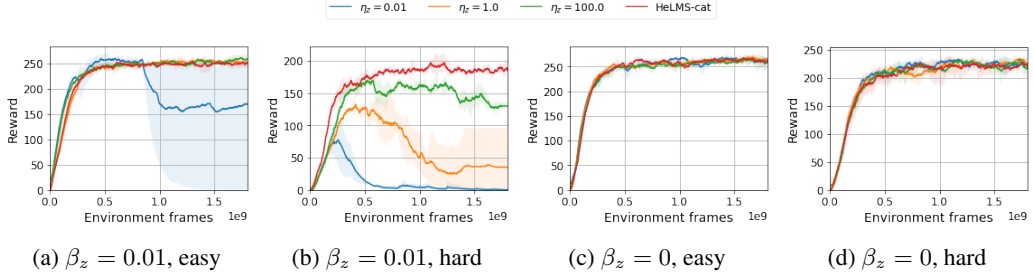

(a) $\beta_z = 0.01$, easy   (b) $\beta_z = 0.01$, hard   (c) $\beta_z = 0$, easy   (d) $\beta_z = 0$, hard

Figure 9: Ablations for KL-regularisation, showing downstream performance with different degrees of online KL-regularisation. Performance is evaluated for easy (object set 4) and hard (object set 3) transfer cases, with sparse staged rewards; when using different offline regularisation coefficient ($\beta_z$) values for the mid-level components.

### A.2 NPMP ABLATION

The Neural Probabilistic Motor Primitives (NPMP) work (Merel et al., 2019) presents a strong baseline approach to learning transferable motor behaviours, and we run ablations to ensure a fair comparison to the strongest possible result. As discussed in the main text, NPMP employs a Gaussian high-level latent encoder with a $AR(1)$ prior in the latent space. We also try a fixed $N(0, I)$ prior (this is equivalent to an $AR(1)$ prior with a coefficient of 0, so can be considered a hyperparameter choice). Since our method benefits from KL-regularisation during RL, we apply this to NPMP as well.

As shown in Figure 10, we find that both changes lead to substantial improvements in the manipulation domain, on all five object sets. Consequently, in our main experiments, we report results with the best variant, using a $N(0, I)$ prior with KL-regularisation during RL.

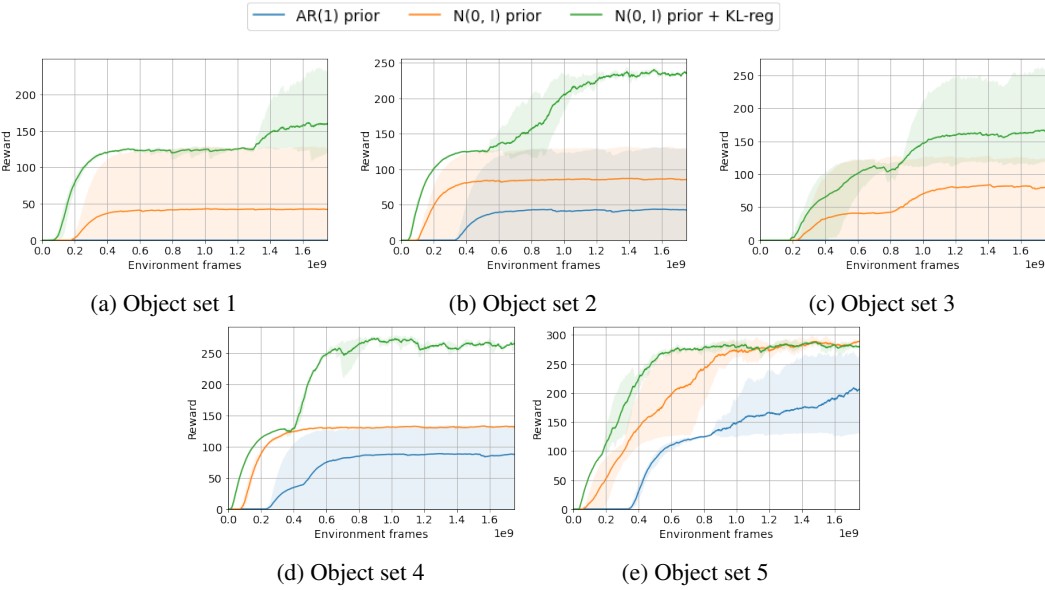

Figure 10: Ablation for NPMP (Merel et al., 2019) using staged sparse reward, with all object sets. We find that using $N(0, I)$ prior with KL-regularisation performs much better in the manipulation domain compared to the original $AR(1)$ prior, so we use this modified baseline.

## B    REINFORCEMENT LEARNING WITH MPO AND RHPO

As discussed in Section 3.2, the hierarchy of skills are transferred to RL in two ways: HeLMS-cat, which learns a new high-level categorical policy $\pi(\mathbf{y}_t \,|\, \mathbf{x}_t)$ via MPO (Abdolmaleki et al., 2018); or HeLMS-mix, which learns a mixture policy $\pi(\mathbf{z}_t \,|\, \mathbf{x}_t) = \sum_{\mathbf{y}_t} \pi(\mathbf{y}_t \,|\, \mathbf{x}_t)\pi(\mathbf{z}_t \,|\, \mathbf{y}_t, \mathbf{x}_t)$ via RHPO (Wulfmeier et al., 2020). We describe the optimisation for both of these cases in the following subsections. For clarity of notation, we omit the additional KL-regularisation terms introduced in Section 3.2 and describe just the base methods of MPO and RHPO when applied to the RL setting in this paper. These KL-terms are incorporated as additional loss terms in the policy improvement stage.

### B.1    HeLMS-CAT VIA MPO

Maximum a posteriori Policy Optimisation (MPO) is an Expectation-Maximisation-based algorithm that performs off-policy updates in three steps: (1) updating the critic; (2) creating a non-parametric intermediate policy by weighting sampled actions using the critic; and (3) updating the parametric policy to fit the critic-reweighted non-parametric policy, with trust region constraints to improve stability. We detail each of these steps below. Note that while the original MPO operates in the environment's action space, we use it here for the high-level controller, to set the categorical variable $\mathbf{y}_t$.

**Policy evaluation**    First, the critic is updated via a TD(0) objective as:

$$\min_\theta L(\theta) = \mathbb{E}_{\mathbf{x}_t, \mathbf{y}_t \sim \mathcal{B}}\Big[\big(Q_\mathrm{T} - Q_\phi(\mathbf{x}_t, \mathbf{y}_t)\big)^2\Big], \qquad (6)$$

Here, $Q_\mathrm{T} = r_t + \gamma \mathbb{E}_{\mathbf{x}_{t+1}, \mathbf{y}_{t+1}}[Q'(s_{t+1}, \mathbf{y}_{t+1})]$ is the 1-step target with the state transition $(\mathbf{x}_t, \mathbf{y}_t, \mathbf{x}_{t+1})$ returned from the replay buffer $\mathcal{B}$, and next action sampled from $\mathbf{y}_{t+1} \sim \pi'(\cdot|\mathbf{x}_{t+1})$. $\pi'$ and $Q'$ are target networks for the policy and the critic, used to stabilise learning.

**Policy improvement** Next, we proceed with the first step of policy improvement by constructing an intermediate non-parametric policy $q(\mathbf{y}_t|\mathbf{x}_t)$, and optimising the following constrained objective:

$$\max_q J(q) = \mathbb{E}_{\mathbf{y}_t \sim q, \mathbf{x}_t \sim \mathcal{B}}\big[Q_\phi(\mathbf{x}_t, \mathbf{y}_t)\big], \ \ \text{s.t. } \mathbb{E}_{\mathbf{x}_t \sim \mathcal{B}}\Big[\text{KL}\big(q(\cdot|\mathbf{x}_t)\|\pi_{\theta_k}(\cdot|\mathbf{x}_t)\big)\Big] \le \epsilon_E, \tag{7}$$

where $\epsilon_E$ defines a bound on the KL divergence between the non-parametric and parametric policies at the current learning step $k$. This constrained optimisation problem has the following closed-form solution:

$$q(\mathbf{y}_t \,|\, \mathbf{x}_t) \propto \pi_{\theta_k}(\mathbf{y}_t \,|\, \mathbf{x}_t) \exp\left(Q_\phi(\mathbf{x}_t, \mathbf{y}_t)/\eta\right). \tag{8}$$

In other words, this step constructs an intermediate policy which reweights samples from the previous policy using exponentiated temperature-scaled critic values. The temperature parameter $\eta$ is derived based on the dual of the Lagrangian; for futher details please refer to (Abdolmaleki et al., 2018).

Finally, we can fit a parametric policy to the non-parametric distribution $q(\mathbf{y}_t \,|\, \mathbf{x}_t)$ by minimising their KL-divergence, subject to a trust-region constraint on the parametric policy:

$$\theta_{k+1} = \arg\min_\theta \mathbb{E}_{\mathbf{x}_t \sim \mathcal{B}}\Big[\text{KL}(q(\mathbf{y}_t \,|\, \mathbf{x}_t) \,\|\, \pi_\theta(\mathbf{y}_t|\mathbf{x}_t))\Big],$$
$$\text{s.t. } \mathbb{E}_{\mathbf{x}_t \sim \mathcal{B}}\Big[\text{KL}\big(\pi_{\theta_{k+1}}(\mathbf{y}_t \,|\, \mathbf{x}_t) \,\|\, \pi_{\theta_k}(\mathbf{y}_t \,|\, \mathbf{x}_t)\big)\Big] \le \epsilon_M. \tag{9}$$

This optimisation problem can be solved via Lagrangian relaxation, with the Lagrangian multiplier $\epsilon_M$ modulating the strength of the trust-region constraint. For further details and full derivations, please refer to (Abdolmaleki et al., 2018).

### B.2 HeLMS-mix via RHPO

RHPO (Wulfmeier et al., 2020) follows a similar optimisation procedure as MPO, but extends it to mixture policies and multi-task settings. We do not exploit the multi-task capability in this work, but utilise RHPO to optimise the mixture policy in latent space, $\pi(\mathbf{z}_t \,|\, \mathbf{x}_t) = \sum_{\mathbf{y}_t} \pi(\mathbf{y}_t \,|\, \mathbf{x}_t) \pi(\mathbf{z}_t \,|\, \mathbf{y}_t, \mathbf{x}_t)$. The Q-function $Q_\phi(\mathbf{x}_t, \mathbf{z}_t)$ and parametric policy $\pi_{\theta_k}(\mathbf{z}_t \,|\, \mathbf{x}_t)$ use the continuous latents $\mathbf{z}_t$ as actions instead of the categorical $\mathbf{y}_t$. This is also in contrast to the original formulation of RHPO, which uses the environment's action space. Compared to MPO, the policy improvement stage of the non-parametric policy is minimally adapted to take into account the new mixture policy. The key difference is in the parametric policy update step, which optimises the following:

$$\theta_{k+1} = \arg\min_\theta \mathbb{E}_{\mathbf{x}_t \sim \mathcal{B}}\Big[\text{KL}(q(\mathbf{z}_t \,|\, \mathbf{x}_t) \,\|\, \pi_\theta(\mathbf{z}_t|\mathbf{x}_t))\Big],$$
$$\text{s.t. } \mathbb{E}_{\mathbf{x}_t \sim \mathcal{B}}\Big[\text{KL}\big(\pi_{\theta_{k+1}}(\mathbf{y}_t \,|\, \mathbf{x}_t) \,\|\, \pi_{\theta_k}(\mathbf{y}_t \,|\, \mathbf{x}_t)\big)$$
$$+ \sum_{\mathbf{y}_t} \text{KL}\big(\pi_{\theta_{k+1}}(\mathbf{z}_t \,|\, \mathbf{y}_t, \mathbf{x}_t) \,\|\, \pi_{\theta_k}(\mathbf{z}_t \,|\, \mathbf{y}_t, \mathbf{x}_t)\big)\Big] \le \epsilon_M. \tag{10}$$

In other words, separate trust-region constraints are applied to a sum of KL-divergences: for the high-level categorical and for each of the mixture components. Following the original RHPO, we separate the single constraint into decoupled constraints that set a different $\epsilon$ for the means, covariances, and categorical ($\epsilon_\mu$, $\epsilon_\sigma$, and $\epsilon_{cat}$, respectively). This allows the optimiser to independently modulate how much the categorical distribution, component means, and component variances can change. For further details and full derivations, please refer to (Wulfmeier et al., 2020).

## C  ELBO DERIVATION AND INTUITIONS

We can compute the Evidence Lower Bound for the state-conditional action distribution, $p(\mathbf{a}_{1:T} \mid \mathbf{x}_{1:T}) \geq ELBO$, as follows:

$$
\begin{aligned}
ELBO &= p(\mathbf{a}_{1:T} \mid \mathbf{x}_{1:T}) - \mathrm{KL}(q(\mathbf{y}_{0:T}, \mathbf{z}_{1:T} \mid \mathbf{x}_{1:T}) \,\|\, p(\mathbf{y}_{0:T}, \mathbf{z}_{1:T} \mid \mathbf{x}_{1:T})) \\
&= \mathbb{E}_{q(\mathbf{y}_{0:T}, \mathbf{z}_{1:T} \mid \mathbf{x}_{1:T})} \Big[ \log p(\mathbf{a}_{1:T}, \mathbf{y}_{0:T}, \mathbf{z}_{1:T} \mid \mathbf{x}_{1:T}) - \log q(\mathbf{y}_{0:T}, \mathbf{z}_{1:T} \mid \mathbf{x}_{1:T}) \Big] \\
&= \mathbb{E}_{q_{1:T}} \Bigg[ \sum_{t=1}^{T} \log p(\mathbf{a}_t \mid \mathbf{z}_t, \mathbf{x}_t) + \log p(\mathbf{z}_t \mid \mathbf{y}_t) + \log p(\mathbf{y}_t \mid \mathbf{y}_{t-1}) \\
&\qquad\qquad - \log q(\mathbf{z}_t \mid \mathbf{y}_t, \mathbf{x}_t) - \log q(\mathbf{y}_t \mid \mathbf{y}_{t-1}, \mathbf{x}) \Bigg] \\
&= \sum_{t=1}^{T} \mathbb{E}_{q_{1:T}} \Bigg[ \log p(\mathbf{a}_t \mid \mathbf{z}_t, \mathbf{x}_t) - \mathrm{KL}(q(\mathbf{z}_t \mid \mathbf{y}_t, \mathbf{x}_t) \,\|\, p(\mathbf{z}_t \mid \mathbf{y}_t)) \\
&\qquad\qquad - \mathrm{KL}(q(\mathbf{y}_t \mid \mathbf{y}_{t-1}, \mathbf{x}_t) \,\|\, p(\mathbf{y}_t \mid \mathbf{y}_{t-1})) \Bigg]
\end{aligned}
\tag{11}
$$

We note that the first two terms in the expectation depend only on timestep $t$, so we can simplify and marginalise exactly over all discrete $\{\mathbf{y}_{1:T}\} \setminus \mathbf{y}_t$. For the final term, we note that the KL at timestep $t$ is constant with respect to $\mathbf{y}_t$ (as it already marginalises over the whole distribution), and only depends on $\mathbf{y}_{t-1}$. Lastly, we will use sampling to approximate the expectation over $\mathbf{z}_t$. This yields the following:

$$
\begin{aligned}
ELBO &= \sum_{t=1}^{T} \mathbb{E}_{q(\mathbf{z}_t \mid \mathbf{y}_t, \mathbf{x}_t)} \Bigg[ \sum_{\mathbf{y}_{0:T}} q(\mathbf{y}_{0:T} \mid \mathbf{x}_{1:T}) \Big( \log p(\mathbf{a}_t \mid \mathbf{z}_t, \mathbf{x}_t) - \mathrm{KL}(q(\mathbf{z}_t \mid \mathbf{y}_t, \mathbf{x}_t) \,\|\, p(\mathbf{z}_t \mid \mathbf{y}_t)) \\
&\qquad\qquad - \mathrm{KL}(q(\mathbf{y}_t \mid \mathbf{y}_{t-1}, \mathbf{x}_t) \,\|\, p(\mathbf{y}_t \mid \mathbf{y}_{t-1})) \Big) \Bigg] \\
ELBO &\approx \sum_{t=1}^{T} \Bigg[ \sum_{\mathbf{y}_t} q(\mathbf{y}_t \mid \mathbf{x}_{1:t}) \Big( \overbrace{\log p(\mathbf{a}_t \mid \tilde{\mathbf{z}}_t^{\{\mathbf{y}_t\}}, \mathbf{x}_t)}^{\text{per-component recon loss}} - \beta_z \overbrace{\mathrm{KL}(q(\mathbf{z}_t \mid \mathbf{y}_t, \mathbf{x}_t) \,\|\, p(\mathbf{z}_t \mid \mathbf{y}_t))}^{\text{per-component KL regulariser}} \Big) \Bigg] \\
&\quad - \beta_y \sum_{t=1}^{T} \Bigg[ \sum_{\mathbf{y}_{t-1}} q(\mathbf{y}_{t-1} \mid \mathbf{x}_{1:t-1}) \underbrace{\mathrm{KL}(q(\mathbf{y}_t \mid \mathbf{y}_{t-1}, \mathbf{x}_t) \,\|\, p(\mathbf{y}_t \mid \mathbf{y}_{t-1}))}_{\text{discrete regulariser}} \Bigg]
\end{aligned}
\tag{12}
$$

where $\tilde{\mathbf{z}}_t^{\{\mathbf{y}_t\}} \sim q(\mathbf{z}_t \mid \mathbf{y}_t, \mathbf{x}_t)$, the coefficients $\beta_y$ and $\beta_z$ can be used to weight the KL terms, and the cumulative component probability $q(\mathbf{y}_t \mid \mathbf{x}_{1:t})$ can be computed iteratively as:

$$
q(\mathbf{y}_t \mid \mathbf{x}_{1:t}) = \sum_{\mathbf{y}_{t-1}} q(\mathbf{y}_t \mid \mathbf{y}_{t-1}, \mathbf{x}_t) q(\mathbf{y}_{t-1} \mid \mathbf{x}_{1:t-1})
\tag{13}
$$

In other words, for each timestep $t$ and each mixture component, we compute the latent sample and the corresponding action log-probability, and the KL-divergence between the component posterior and prior. This is then marginalised over all $\mathbf{y}_t$, with an additional KL over the categorical transitions.

Structuring the graphical model and ELBO in this form has a number of useful properties. First, the ELBO terms include an action reconstruction loss and KL term for each mixture component, scaled by the posterior probability of each component given the history. For a given state, this pressures the model to assign higher posterior probability to components that have low reconstruction cost or KL, which allows different components to specialise for different parts of the state space. Second, the categorical KL between posterior and prior categorical transition distributions is scaled by the

posterior probability of the previous component given history $q(\mathbf{y}_{t-1} \,|\, \mathbf{x}_{1:t-1})$: this allows the relative probabilities of past skill transitions along a trajectory to be considered when regularising the current skill distribution. Finally, this formulation does not require any sampling or backpropagation through the categorical variable: starting from $t = 0$, the terms for each timestep can be efficiently computed by recursively updating the posterior over components given history ($q(\mathbf{y}_t \,|\, \mathbf{x}_{1:t})$), and summing over all possible categorical values at each timestep.

## D  ENVIRONMENT PARAMETERS

As discussed earlier in the paper, all experiments take place in a MuJoCo-based object manipulation environment using a Sawyer robot manipulator and three objects: red, green, and blue. The state variables in the Sawyer environment are shown in Table 3. All state variables are stacked for 3 frames for all agents. The object states are only provided to the mid-level and high-level for HeLMS runs, and the camera images are only used by the high- and mid-level controller in the vision transfer experiments (without object states).

The action space is also shown in Table 4. Since the action dimensions vary significantly in range, they are normalised to be between $[-1, 1]$ for all methods during learning.

When learning via RL, we apply domain randomisation to physics (but not visual randomisation), and a randomly sampled action delay of 0-2 timesteps. This is applied for all approaches, and ensures that we can learn a policy that is robust to small changes in the environment.

| Proprioception | |
|---|---|
| **State** | **Dims** |
| Joint angles | 7 |
| Joint velocities | 7 |
| Joint torque | 7 |
| TCP pose | 7 |
| TCP velocity | 6 |
| Wrist angle | 1 |
| Wrist velocity | 1 |
| Wrist force | 3 |
| Wrist torque | 3 |
| Binary grasp sensor | 1 |

| Object states | |
|---|---|
| **State** | **Dims** |
| Absolute pose (red) | 7 |
| Absolute pose (green) | 7 |
| Absolute pose (blue) | 7 |
| Distance to pinch (red) | 7 |
| Distance to pinch (green) | 7 |
| Distance to pinch (blue) | 7 |

| Vision | |
|---|---|
| **State** | **Dims** |
| Camera images | $64 \times 64 \times 3$ |

Table 3: Details of state variables used by the agent.

| Action | Dims | Range |
|---|---|---|
| Gripper translational velocity (x-y-z) | 3 | $[-0.07, 0.07]$ m/s |
| Wrist rotation velocity | 1 | $[-1, 1]$ rad/s |
| Finger speed | 1 | $[-255, 255]$ tics/s |

Table 4: Action space details for the Sawyer environment.

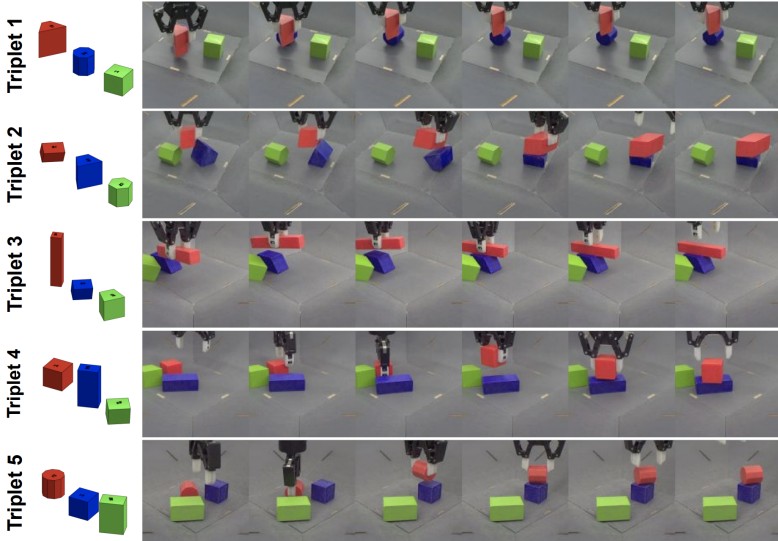

Figure 11: The five object sets (triplets) used in the paper. This image has been taken directly from (Lee et al., 2021) for clarity.

### D.1 OBJECT SETS

As discussed in the main paper, we use the object sets defined by Lee et al. (2021), which are carefully designed to cover different object geometries and affordances, presenting different challenges for object interaction tasks. The object sets are shown in Figure 11 (the image has been taken directly from (Lee et al., 2021) for clarity), and feature both simulated and real-world versions; in this paper we focus on the simulated versions. As discussed in detail by (Lee et al., 2021), each object set has a different degree of difficulty and presents a different challenge to the task of stacking red-on-blue:

- In object set 1, the red object has slanted surfaces that make it difficult to grasp, while the blue object is an octagonal prism that can roll.
- In object set 2, the blue object has slanted surfaces, such that the red object will likely slide off unless the blue object is first reoriented.
- In object set 3, the red object is long and narrow, requiring a precise grasp and careful placement.
- Object set 4 is the easiest case with rectangular prisms for both red and blue.
- Object set 5 is also relatively easy, but the blue object has ten faces, meaning limited surface area for stacking.

For more details about the object sets and the rationale behind their design, we refer the reader to (Lee et al., 2021).

## E NETWORK ARCHITECTURES AND HYPERPARAMETERS

The network architecture details and hyperparameters for HeLMS are shown in Table 5. Parameter sweeps were performed for the $\beta$ coefficients during offline learning and the $\eta$ coefficients during RL. Small sweeps were also performed for the RHPO $\epsilon$ parameters (refer to (Wulfmeier et al., 2020) for details), but these were found to be fairly insensitive. All other parameters were kept fixed, and used for all methods except where highlighted in the following subsections. All RL experiments were run with 3 seeds to capture variation in each method.

For network architectures, all experiments except for vision used simple 2-layer MLPs for the high- and low-level controllers, and for each mid-level mixture component. An input representation network was used to encode the inputs before passing them to the networks that were learned from scratch: i.e. the high-level for state-based experiments, and both high- and mid-level for vision (re-

call that while the state-based experiments can reuse the mid-level components conditioned on object state, the vision-based policy learned them from scratch and KL-regularised to the offline mid-level skills). The critic network was a 3-layer MLP, applied to the output of another input representation network (separate to the actor, but with the same architecture) with concatenated action.

**Offline learning parameters**

| Name | Value |
|---|---|
| Latent space dimension | 8 |
| Number of mid-level components, $K$ | 5 for `red_on_blue` data, 10 for `all_pairs` data |
| Low-level network | 2-hidden layer MLP, $\{256, 256\}$ units |
| Low-level head | Gaussian, tanh-on-mean, fixed $\sigma = 0.1$ |
| Mid-level network | 2-hidden layer MLP for each component, $\{256, 256\}$ units |
| Mid-level head | Gaussian, learned $\sigma \in [0.01, 1.0]$ |
| High-level network | 2-hidden layer MLP, $\{256, 256\}$ units |
| High-level head | $K$-way softmax |
| Activation function | elu |
| Encoder look-ahead duration | 5 timesteps |
| $\beta_y$ | 1.0 |
| $\beta_z$ | 0.1, 0.0 (object generalisation) |
| Batch size | 128 |
| Learning rate | $10^{-4}$ |
| Dataset trajectory length | 25 |

**Online RL parameters**

| Name | Value |
|---|---|
| Number of seeds | 3 (all experiments) |
| Input representation network (state) | Input normalizer layer (linear layer with 256 units, layer-norm, and tanh-on-output) |
| High-level network (state) | 2-hidden layer MLP, $\{256, 256\}$ units |
| Input representation network (vision) | MLP on proprio and ResNet with three layers of $\{2, 2, 2\}$ blocks corresponding to $\{32, 64, 128\}$ channels |
| High-level network (vision) | 2-hidden layer MLP, $\{256, 256\}$ units |
| Mid-level network (vision) | 2-hidden layer MLP for each component, $\{256, 256\}$ units |
| Critic network | 3-hidden layer MLP, $\{256, 256, 256\}$ units with RNN |
| Activation function | elu |
| $\eta_y$ | 0.1 (vision), 0.01 |
| $\eta_z$ | 0.1 (pyramid and vision), 0.01 (object generalisation) |
| Number of actors | 1500 |
| Batch size | 512 |
| Trajectory length | 10 |
| Learning rate | $2 \times 10^{-4}$ |
| Number of action samples | 20 |
| RHPO categorical constraint $\epsilon_{cat}$ | 1.0 |
| RHPO mean constraint $\epsilon_{\mu}$ | $5 \times 10^{-3}$ |
| RHPO covariance constraint $\epsilon_{\sigma}$ | $10^{-4}$ |

Table 5: Hyperparameters and architecture details for HeLMS, for both offline training and RL.

## F  REWARDS

Throughout the experiments, we employ different reward functions for different tasks and to study the efficacy of our method in sparse versus dense reward scenarios.

**Reward stages and primitive functions**    The reward functions for stacking and pyramid tasks use various reward primitives and staged rewards for completing sub-tasks. Each of these rewards are within the range of $[0, 1]$

These include:

- `reach(obj)`: a shaped distance reward to bring the TCP to within a certain tolerance of `obj`.
- `grasp()`: a binary reward for triggering the gripper's grasp sensor.
- `close_fingers()`: a shaped distance reward to bring the fingers inwards.
- `lift(obj)`: shaped reward for lifting the gripper sufficiently high above `obj`.
- `hover(obj1,obj2)`: shaped reward for holding `obj1` above `obj2`.
- `stack(obj1,obj2)`: a sparse reward, only provided if `obj1` is on top of `obj2` to within both a horizontal and vertical tolerance.
- `above(obj,dist)`: shaped reward for being `dist` above `obj`, but anywhere horizontally.
- `pyramid(obj1,obj2,obj3)`: a sparse reward, only provided if `obj3` is on top of the point midway between `obj1` and `obj2`, to within both a horizontal and vertical tolerance.
- `place_near(obj1,obj2)`: sparse reward provided if `obj1` is sufficiently near `obj2`.

**Dense stacking reward**   The dense stacking reward contains a number of stages, where each stage represents a sub-task and has a maximum reward of 1. The stages are:

- `reach(red) AND grasp()`: Reach and grasp the red object.
- `lift(red) AND grasp()`: Lift the red object.
- `hover(red,blue)`: Hover with the red object above the blue object.
- `stack(red,blue)`: Place the red object on top of the blue one.
- `stack(red,blue) AND above(red)`: Move the gripper above after a completed stack.

At each timestep, the latest stage to receive non-zero reward is considered to be the current stage, and all previous stages are assigned a reward of 1. The reward for this timestep is then obtained by summing rewards for all stages, and scaling by the number of stages, to ensure the highest possible reward on any timestep is 1.

**Sparse staged stacking reward**   The sparse staged stacking reward is similar to the dense reward variant, but each stage is sparsified by only providing the reward for the stage once it exceeds a value of 0.95.

This scenario emulates an important real-world problem: that it may be difficult in certain cases to specify carefully shaped meaningful rewards, and it can often be easier to specify (sparsely) whether a condition (such as stacking) has been met.

**Sparse stacking reward**   This fully sparse reward uses the `stack(red,blue)` function to provide reward only when conditions for stacking red on blue have been met.

**Pyramid reward**   The pyramid-building reward uses a staged sparse reward, where each stage represents a sub-task and has a maximum reward of 1. If a stage has dense reward, it is sparsified by only providing the reward once it exceeds a value of 0.95. The stages are:

- `reach(red) AND grasp()`: Reach and grasp the red object.
- `lift(red) AND grasp()`: Lift the red object.
- `hover(red,green)`: Hover with the red object above the green object (with a larger horizontal tolerance, as it does not need to be directly above).
- `place_near(red,green)`: Place the red object sufficiently close to the green object.
- `reach(blue) AND grasp()`: Reach and grasp the blue object.
- `lift(blue) AND grasp()`: Lift the blue object.
- `hover(blue,green) AND hover(blue,red)`: Hover with the blue object above the central position between red and green objects.
- `pyramid(blue,red,green)`: Place the blue object on top to make a pyramid.
- `pyramid(blue,red,green) AND above(blue)`: Move the gripper above after a completed stack.

At each timestep, the latest stage to receive non-zero reward is considered to be the current stage, and all previous stages are assigned a reward of 1. The reward for this timestep is then obtained by summing rewards for all stages, and scaling by the number of stages, to ensure the highest possible reward on any timestep is 1.

