# OpenReview forum: "Learning transferable motor skills with hierarchical latent mixture policies"
_ICLR.cc/2022/Conference — ICLR 2022 Spotlight_

### Official Review · Reviewer_cfLn · 2021-10-30

**Correctness:** 3
**Technical Novelty And Significance:** 3
**Empirical Novelty And Significance:** 3
**Recommendation:** 8
**Confidence:** 4

**Main Review:**

STRENGTH:

S1: The paper considers a very important problem.

I think that skill extrapolation and reuse it is at the core of providing more sample efficient reinforcement learning and to allow its application to real-world tasks.

S2.1: I think that the proposed method is sound.

The idea of breaking the three-level of hierarchy makes a lot of sense. Many existing approaches in fact propose a similar structure, which, in higher detail, can be described as a mixture of parametric skills.

The idea of incorporating information asymmetries, as also noted by the authors, was empirically shown to be also effective.

S2.2: Many works I am aware of, have limited applicability, since they do not make use of neural networks, and they do not scale well with high-dimensional input. This work does not suffer from this problem. Furthermore, the extracted skills can be easily used in many classic RL algorithms.

S3. The experimental section consistently supports the statement made, showing the efficacy of the proposed method. I especially enjoyed the ablation study.

WEAKNESSES

W1: The readability of this paper can, in my opinion, be improved.

In particular:

W1.1: the paragraph "Training via the Evidence Lower bound" is a bit messy. The quantities $p_{\psi_{y,a}}$, $q_{\phi_{y,z}}$, etc, have not been defined before. I think that these models should have been defined at the beginning of  2.1.
Furthermore, the left-hand side of Equation 3 could have been defined explicitly, even though that the ELBO is a widely known concept.

W1.2: Equations 4 and 5 present just RL objectives. I think that the authors could have spent some lines to write how MPO optimizes these objectives. Writing down the optimization process will explicitly clarify which parameters will be updated. The current form of Section 2.3. is, in my opinion, too fuzzy.

W1.3: I think that many equations contain some typos. For example,

Equation 1: on the right side we have p(y_0), but why on the right-hand-side $p(y_{1:T}, Z_{1:T})$ does not take in account of $p(y_0)$?
The same questions arise also in Equation 2.

The summations $\sum_{y_{t-1}}$ in the derivation of Equation 3 should be better defined.

In Equations 4 and 5, $x_t$ is outside the expectation. This is wrong. Question: shouldn't the KL also be discounted with \gamma^t? If the KL is summed for each $t$ term from $0$ to $\infty$, then the summation will be unbounded (and, therefore, impossible to optimize).

W2.

The proposed method, in my understanding, can be very much regarded as "skills segmentation" "movement primitive segmentation", [1-4]...

Furthermore, the mixture of skills (or movement primitives) has been also discussed widely [5, 6, 7, 9, 10]. Notice that the probabilistic model of [6, 10] resembles closely your approach (discrete selection of parametric skills).
The mixture of movement primitives has also been utilized in reinforcement learning, leading to a very similar probabilistic model (with a discrete variable selecting the skill, and the continuous variable selecting the skill's parameters) [7-10].

If you dig a bit, you will find many more connections to your current work.

I am also aware of this very recent work, which also uses high-level skills to perform RL tasks [11]. Their motivation and result are, in my opinion, strongly supporting the evidence of your result, although (in my opinion) your work is more complete.

W3: as described above, and also as partially acknowledged in your paper, people have been working on this topic. I like how you have put things together, and I think your work is promising. I think that the contribution is enough, but not particularly strong.

To summarize, I think that the idea behind your method is good. I find it a pity that it misses a better related-work contextualization, mathematical precision, and exposition clarity.

QUESTIONS:

- Can the authors clarify my doubts about the equations?
- Can the author draw a link between their work and HMM?

[1] Scott Niekum, Sarah Osentoski, George Konidaris, Andrew G. Barto. Learning and Generalization of Complex Tasks
from Unstructured Demonstrations. IROS 2012.
[2] Rudolf Lioutikov, Gerhard Neumann, Guilherme Maeda, Jan Peters. Probabilistic Segmentation Applied to an Assembly Task. Humanoids 2015.
[3] Rudolf Lioutikov, Gerhard Neumann, Guilherme Maeda, Jan Peters., Learning movement primitive libraries
through probabilistic segmentation, IJRR 2017.
[4] Bjorn Kruger, Anna Vogele, Tobias Willig, Angela Yao, Reinhard Klein, Efficient Unsupervised Temporal
Segmentation of Motion Data.
[5]  Affan Pervez,  Dongheui Lee. Learning task-parameterized dynamic movement primitives using a mixture of GMMs.
[6] Elmar Rueckert, Jan Mundo, Alexandros Paraschos, Jan Peters, and Gerhard Neumann. Extracting Low-Dimensional Control Variables for Movement Primitives. ICRA 2015.
[7] Katharina Muelling, Jens Kober, Jan Peters. Learning Table Tennis with a Mixture of Motor Primitives. Humanoids 2010
[8] Katharina Mülling, Jens Kober, Oliver Kroemer and Jan Peters. Learning to select and generalize striking movements in robot table tennis. IJRR
[9] Adria Colome, and Carme Torras, Dimensionality Reduction in Learning Gaussian Mixture Models of Movement Primitives for Contextualized Action Selection and Adaptation. RAL 2018.
[10] Samuele Tosatto, Georgia Chalvatzaki, and Jan Peters. Contextual Latent-Movements Off-Policy Optimization for Robotic Manipulation Skills. ICRA 2021.
[11] Dalal Murtaza, Pathak Deepak and Salakhutdinov Ruslan. Accelerating Robotic Reinforcement Learning via Parameterized Action Primitives. NeurIPS 2021.


**Summary Of The Paper:**

This paper presents a hierarchical model for skills extraction from an online dataset. In particular, the model can be seen as a 3-level hierarchy. The highest level of the hierarchy selects a discrete variable that corresponds to the index of a skill. The mid-level selects the continuous parameter that defines the skill. The lower level executes the skill.

The considered problem is crucial in reinforcement learning. In particular, the definition of skills to be reused across different tasks is essential to achieve better sample efficiency, and crucial to allow reinforcement learning to be applied directly on challenging real robotic tasks.

When dealing with an off-line dataset, one wants potentially to first extract a set of skills (say, low-level policies) and then to learn via RL the selector of such policies. In this way, one both simplifies the problem (increasing sample efficiency) and allows the skills to be reused in different tasks.

The authors propose a graphical model that consists, per time step, of four variables: $x_t$, $y_t$ $z_t$ and $a_t$. $x_t$ represents the MDP's state, $y_t$ is the discrete selector of the skill, $z_t$ the (continuous) parameters of the skill, and $a_t$ is the output at time $t$ of the selected skill (and can represent, for example, the joint velocities of the robot).

After learning via variational inference the probabilistic model, the authors propose to refine the learned behavior via MPO, learning the skill selector $p(y_t | y_{t-1},x_t)$, the skill parameters $p(z_t | y_t, x_t)$, and the skill p(a_t | z_t, y_t).

Further, the authors propose 1) to use an asymmetric information approach, where each level of the described hierarchy sees a different definition of the state. In particular, the lower layer sees the robotic state (proprioception), the mid-layer sees the position of objects in the scene, and the highest level sees both the information 2) the models use "gated"-heads: the categorical variable $y_t$ is fed to the last part of the neural network.

Once the hierarchy has been learned, one can tune it on the same or different tasks. In particular, the lowest layer is kept fixed, while only the highest, or the highest, and the mid-layer are learned.

The authors propose to use MPO as a reinforcement learning algorithm. They perform a few ablation studies and experiments to empirically prove the efficacy of their method.

**Summary Of The Review:**

STRENGTH

1. The paper considers a very important problem (skills extraction and reuse).
2. The proposed method is sound. Furthermore, the employment of variational inference with deep NN allows the method to be scalable and usable on high-dimensional tasks.
3. The experiment section is well structured and supports nicely the statements made.

WEAKNESSES

1. I think the paper is not very clear. In particular, some passages are missing, and some equations seem not to be correct.
2. I think that a large part of related work is missing, especially work prior to 2017.

I would give 8 (Accept, Good Paper), but I really think that this paper should be organized better, and the equations are not clear. I have an issue with every single equation in the paper. May I ask the authors to clarify my doubts?



================ SUMMARY OF CHANGES ======================

My main concerns were mainly about the clarity of the paper and the lack of a large body of related work.
I think that the authors addressed both the issues in their new version of the paper.

The new paper is much clearer, and the proper discussion of related work helps in defining more rigorously what was the contribution of this paper. Therefore, I increased my score from 6 to 8.

---

> ### Author Response · Authors · 2021-11-19
> **Authors' response to Reviewer cfLn**
>
> Thanks for your rigorous and detailed comments. We have carefully addressed all of the concerns you have raised, by expanding the explanation and justification for the ELBO, fixing all equations, and significantly extending the citations and related work. We elaborate on particular questions and comments below.
>
> **The paragraph "Training via the Evidence Lower bound" is a bit messy. The quantities [phi and psi] have not been defined before. I think that these models should have been defined at the beginning of 2.1. Furthermore, the left-hand side of Equation 3 could have been defined explicitly, even though the ELBO is a widely known concept.**
>
> We have changed the paper to define the trainable parameters more explicitly, rather than introducing them in the equation directly. To avoid cluttered notation early on, we mention them first in the “training via the ELBO” subsection. We also added an additional explicit definition of the ELBO, and further discussion and intuitions - due to space constraints, we reserve these for the derivation in the appendix.
>
> **Equations 4 and 5 present just RL objectives. I think that the authors could have spent some lines to write how MPO optimizes these objectives. Writing down the optimization process will explicitly clarify which parameters will be updated. The current form of Section 2.3. is, in my opinion, too fuzzy.**
>
> We have added details of the optimization procedures for MPO and RHPO in the appendix, which we hope will also clarify the losses used and which parameters are updated.
>
> **Typos in equations [y_0, summations over y, and KL terms during RL]**
>
> Thank you for these detailed comments. We have rectified all of these issues: the joint distributions over y and z now include y_0, the summations have been made more clear by specifying the possible values of y upfront, and the KL terms in the RL equations have been fixed and appropriately discounted.
>
> **[Related work in segmentation of skills or movement primitives]**
>
> Thank you for making us aware of this extensive list of existing work. We have added a paragraph in the related work section to capture the literature on unsupervised skill segmentation, and movement primitives, including the works on mixtures of parametric skills; and incorporated the remaining references into the existing text as well.
>
> **Can the author draw a link between their work and HMM?**
>
> Our work is related to HMM in that it models a trajectory sequence via a sequence of [discrete and continuous] hidden variables. More specifically, the prior sequence of continuous latent variables z_1:T are conditionally independent given the categorical latents, y_1:T, and the prior for each categorical latent y_t is conditioned on y_{t-1}.  In other words, the prior over the sequence of continuous latent variables can be considered a HMM with a Gaussian emission distribution. We have now clarified this in the paper.

---

> > ### Comment · Reviewer_cfLn · 2021-11-21
> > **Answer to the Rebuttal**
> >
> > Dear authors,
> >
> > I want to acknowledge that I carefully read 1. your response, 2. the new version of the paper, and 3. the other reviews.
> > I am satisfied with your answers, and with the changes made to the paper. I do not have further questions.
> >
> > I will take these inputs into consideration for my final evaluation.
> >
> > P.S.
> >
> > I still think there is a very minor typo in Equation 4.
> >
> > $$\mathbb{E}_{\pi(y_t | x_t)}\left[ \sum_t \dots \right].$$
> >
> > $t$ cannot be at the same time outside the expectation and iterating variable of the summation. I suggest either using a bit imprecise but understandable notation like in Equation 5
> >
> > $$\mathbb{E}_{\pi}\left[ \sum_t \dots \right].$$
> >
> > Or better, to introduce the concept of trajectory $\tau_\pi = \\{(x_t, y_t, z_t, r_t)\\}_{t=0}^T$ and
> >
> > $$\mathbb{E}_{\tau_\pi}\left[ \sum_t \dots \right].$$
> >
> > taking in this way average w.r.t. all the stochastic variables that frame your problem. Personally, I would use this second approach both for Eq. 4 and Eq. 5. Nevertheless, I recognize that these are very small details.
> >
> >
> > Best regards.

---

> > > ### Author Response · Authors · 2021-11-22
> > > **We have fixed the remaining typo**
> > >
> > > Thank you for your response and the additional detailed suggestions. We have fixed this typo in the updated revision.

---

### Official Review · Reviewer_sbA9 · 2021-10-31

**Correctness:** 3
**Technical Novelty And Significance:** 2
**Empirical Novelty And Significance:** 3
**Recommendation:** 8
**Confidence:** 4

**Main Review:**

The most interesting aspect of the proposed system is the fact that transfer by retraining only at the discrete level can be so advantageous. This is true at least as long as you stick to manipulation for which skills like moving the end-effector and opening/closing the gripper are quite generally applicable. However, since different levels may work at different dimensions of the state space, it might be possible to retrain the mid-level controller from one modality to another, something that is illustrated in the experiments.

The experimental section is quite extensive and includes comparisons to other alternative methods, as well as ablation studies. Two variants are tested for transfer learning; one that only retrains the high-level controller (HeLMS-cat) and one that also retrains the mid-level (HeLMS-mix), while the low-level controller is always kept fixed. The first option seems to be more beneficial in complex cases with sparse rewards. In fact, it is a bit surprising that HeLMS-mix rarely seems to surpass HeLMS-cat. It is easy to assume that this is due to the similarity between manipulation tasks when it comes to what skill sets are required.

Compared to the other alternative methods tested, HeLMS seems to cover a larger state space during exploration, which in turn leads to higher rewards, when rewards are sparse. With the three-level hierarchy, exploration can be done at a higher level of abstraction while exploiting skills already learned.

There are some questions worth asking regarding the experimental results. It is shown that for more complicated tasks, HeLMS-mix might come to the point where the average reward starts to decrease. It would seem more reasonable if the reward had just flattened out at a lower level, which happens in most other cases. An explanation given in the paper is that in such cases spurious reward correlations might cause the skills to drift. Does this instability come as a result of including an additional hierarchical level, since none of the other tested methods seems to show anything similar? Another question is why NPMP performs better on object set 2 specifically? Is there any reasonable explanation or is it just a coincidence?

When referring to HeLMS in the experiments, it would be good if the paper mentioned either HeLMS-cat or HeLMS-mix, unless both variants are intended. Otherwise, the reader has to go into the text to see which variant was actually used. For example, which variant is used in Table I? By the way, in this table, what is going on with the space coverage in the dimension of grasping? Since it is binary it looks as if none of the other methods really tries to explore grasping, which is odd given the nature of the task given.

Figures 7 c)-d) use the wrong notation for the regularization weight, at least compared to what is written in the text. Also, instead of HeLMS-cat the label says just HeLM.

On page three, “ELBO <= p(a|x)” is written in the wrong order. It is correct in appendix B. Other than that, the paper is very easy to read and understand, with clarity in both language and notations.


**Summary Of The Paper:**

This paper proposes a three-level hierarchy for learning motor skills. At the bottom, there is a low-level controller for action generation and at the top a high-level controller that generates sequences of skills, with continuous control signals generated by skills at the mid-level. When transferring motor skills from one task to another using reinforcement learning, it is possible to either retrain only the high-level controller or the mid-level one as well. In experiments, it is shown that for transfers to more complex manipulation tasks with sparse rewards, you are more likely to get convergence in training with positive rewards, if only retraining the high-level controller. Thus you are given more flexibility and are possibly able to learn more complex tasks, with the proposed hierarchy compared to earlier methods based on only two levels.

**Summary Of The Review:**

The proposed three-level hierarchy for learning motor skills might not be groundbreaking in itself, but it brings some interesting advantages and flexibility that is illustrated in the paper. It is interesting to see how far you can go in complicated cases by reusing an already learned skill set. At the same time, you might be able to achieve higher rewards in the end if skills are also refined. For that reason, this paper is definitely worth reading. It is possible to imagine training schemes in which you gradually reduce the regularization of the mid-level skills to make them more specific at the end of training, and thus possibly get the best of both worlds.

---

> ### Author Response · Authors · 2021-11-19
> **Authors' response to Reviewer sbA9**
>
> Thank you for your useful feedback; we have made several improvements to the paper, including additional analysis and discussion for the different experiments. We address your concerns in detail below.
>
> **There are some questions worth asking regarding the experimental results. It is shown that for more complicated tasks, HeLMS-mix might come to the point where the average reward starts to decrease. It would seem more reasonable if the reward had just flattened out at a lower level, which happens in most other cases. An explanation given in the paper is that in such cases spurious reward correlations might cause the skills to drift. Does this instability come as a result of including an additional hierarchical level, since none of the other tested methods seems to show anything similar?**
>
> This is a great question. We have some more recent experiments with better results for HeLMS-mix when removing the regularisation to the N(0,I) prior during the offline learning phase. In this case, it appears that transfer performance is not sensitive to KL-regularisation during RL, which suggests a more interesting and nuanced observation: that when the mid-level skills have each been regularised towards a N(0, I) prior when being learned offline, they need to be strongly KL-regularised during transfer to more challenging objects. We conjecture that with a skill space regularised to N(0, I), the different mid-level skills are closer together, such that KL-regularisation in RL is necessary for a skill to avoid drifting and overlapping with other skills (ie. skill degeneracy).
> We believe this is an interesting observation, and have added more analysis of this in the appendix. We will also re-run experiments with learned p(z|y) priors to provide a full ablation showing how the dependence on KL-regularisation during RL varies based on the offline prior.
>
> **Another question is why NPMP performs better on object set 2 specifically? Is there any reasonable explanation or is it just a coincidence?**
>
> One intuition also shared in prior work is that hierarchy is a tradeoff, incorporating additional structure to aid exploration / search, but potentially at the cost of flexibility or generality. A key property of object set 2 is that the bottom object has slanted surfaces and often needs to be re-oriented. Thus, we conjecture that in this case, because of the uncommon affordance required, the two-level hierarchy of NPMP can perform better, even if it is much slower to learn the solution. We have added more discussion of this in the paper, as well as an avenue for future work: to incorporate the additional high-level behaviour abstractions of HeLMS while maintaining the flexibility of the lower levels.
>
> **When referring to HeLMS in the experiments, it would be good if the paper mentioned either HeLMS-cat or HeLMS-mix, unless both variants are intended. Otherwise, the reader has to go into the text to see which variant was actually used. For example, which variant is used in Table I?**
>
> Thanks for the note - we have made sure to now specify which variant is used in each case. For the Table, both HeLMS-cat and HeLMS-mix are identical (i.e. this is at the start of reinforcement learning, with the transferred mid-level skills and a randomly initialised categorical controller). We have made this more clear in the text.
>
> **In this table, what is going on with the space coverage in the dimension of grasping? Since it is binary it looks as if none of the other methods really tries to explore grasping, which is odd given the nature of the task given.**
>
> The results in the table are at the start of reinforcement learning (without further training), and are accumulated over 1000 episodes. Intuitively, initial exploration with from-scratch techniques tend to be more random arm movements, while HeLMS acts over the mid-level skills, which are shown in Section 4.1 to represent more abstract, directed behaviours like reaching towards an object, or closing the gripper for a grasp. Note that the BC methods also explore grasping behaviours, as is to be expected in this setting, but can struggle to generalise zero-shot when transferring to RL, such that only some episodes reach the grasping stage. We have changed the paper to try and clarify this intuition.
>
> **[Other comments on notation, equations, and labels]**
>
> These have now been fixed in the paper.

---

> > ### Comment · Reviewer_sbA9 · 2021-11-22
> > **Answer to the Rebuttal**
> >
> > That you for your answers and the extensive changes made to the paper. The proposed motivation for the observed decrease in the average reward is reasonable. This is something worth further investigation beyond the scope of this paper.

---

### Official Review · Reviewer_NRgW · 2021-10-31

**Correctness:** 3
**Technical Novelty And Significance:** 2
**Empirical Novelty And Significance:** 3
**Recommendation:** 8
**Confidence:** 4

**Main Review:**

Strengths:

Hierarchical policies are a great idea, and the proposed model seems like a sensible approach to integrate both discrete and continuous latent variables into a policy, in contrast to many existing approaches which use discrete latent indicator variables to trigger fixed policies.

Results show the proposed architecture seems to work, and I was particularly interested in the comparison with a hierarchical behaviour cloning model.

However, the most interesting part of this work lies around the RL phase of the work, where different levels of mixing are explored (completely freezing skills, vs allowing greater skill adaptation. This area is of particular interest as we move beyond "look we discovered some skills and re-used them" to more realistic online learning settings.

Weaknesses:

The paper is in some need of smoothing, and I found this a difficult read, despite being familiar with the field. In particular, I believe the paper would benefit from a clearer problem formulation (ie. is the problem setting a two stage learning from demonstration one followed by an RL phase that makes use of the initial learned skills, or is it a tabular rasa setting with skills gradually incorporated and re-used) Initial sections suggest the former, but some experimental settings and baselines seem to indicate both. I understand that the architecture is very general, and the aim was to show it can be used in both settings with tweaks, but I think the paper would benefit greatly from making the different settings more clear in an expanded problem formulation.

I would also greatly appreciate a more structured experiments setting, more clearly delineating the tasks and with a much more focused aim. As an example, the baselines are all designed to solve slightly different problems, and the broad set of experiments are not necessarily presented to make a cohesive argument, rather they seem to aim to answer many separate questions. A potential option would be to restructure this section around the question of how much prior do we need and how quickly we can allow skill adaptation, which is a very interesting research question.

Questions/ Comments:

There is a wealth of literature in hierachical and resuable skills discovery for RL/ continuous control that is missing from this work, in particular around switching nonlinear dynamical systems. See below for a small selection of work deserving mention.

This paper is particularly important, as it discusses a range of architectures to embed discrete latent variables for skill discovery and later re-use: Carlos Florensa, Yan Duan, Pieter Abbeel, "Stochastic Neural Networks for Hierarchical Reinforcement Learning" ICLR 2017

Other work on re-usable skill discovery using discrete latent variable models:
Scott Niekum, Andrew Barto, "Clustering via Dirichlet Process Mixture Models for Portable Skill Discovery", Neurips 2011

P. Ranchod, B. Rosman and G. Konidaris, "Nonparametric Bayesian reward segmentation for skill discovery using inverse reinforcement learning," 2015 IEEE/RSJ International Conference on Intelligent Robots and Systems (IROS), 2015, pp. 471-477, doi: 10.1109/IROS.2015.7353414.

Kipf, Thomas, et al. "Compile: Compositional imitation learning and execution." International Conference on Machine Learning. PMLR, 2019.

Hany Abdulsamad, Jan Peters, "Hierarchical Decomposition of Nonlinear Dynamics and Control for System Identification and Policy Distillation", Proceedings of the 2nd Conference on Learning for Dynamics and Control, PMLR 120:904-914, 2020.

D. Tanneberg, K. Ploeger, E. Rueckert and J. Peters, "SKID RAW: Skill Discovery From Raw Trajectories," in IEEE Robotics and Automation Letters, vol. 6, no. 3, pp. 4696-4703, July 2021, doi: 10.1109/LRA.2021.3068891.

Categorical regularisation in hierarchical models has been proposed previously in:
Michael Burke, Yordan Hristov, Subramanian Ramamoorthy, "Hybrid system identification using switching density networks", Proceedings of the Conference on Robot Learning, PMLR 100:172-181, 2020.

See also:
Zhe Dong, Bryan Seybold, Kevin Murphy, Hung Bui, "Collapsed Amortized Variational Inference for Switching Nonlinear Dynamical Systems", Proceedings of the 37th International Conference on Machine Learning, PMLR 119:2638-2647, 2020.

Minor queries/ comments:
I assume that the number of skills needs to be pre-specified and remains fixed?

Section 2.3: "Following most previous work, we ..." - citation needed.


**Summary Of The Paper:**

This paper introduces a latent variable controller model that allows for reusable skill learning in behaviour cloning and reinforcement learning settings. The architecture comprises three stages or levels. At the highest level, input state information (eg. proprioception, visual input, object state information) is passed through an MLP to produce a discrete latent state skill selection variable, incorporating a discrete latent transition model. This skill selection prior is then used in a mid level network operating on the same input information, to produce a continuous latent state, conditioned on this discrete latent variable. Finally, this latent state, together with the input state is used by an actor network to produce actions.

The model is trained used a heuristic process depending on the deployment setting, with different elements frozen at different times, or through the use of a mixture agent that can be initialised from scratch or using previous skills (a similar approach is explored by Florensa et al., ICLR 2017). A KL regularisation on the discrete latent variable skill selector is used (also in prior work - Burke et al., CoRL 2019). Results show that this approach is effective on a range of tasks, and that skill re-use (as expected) out performs learning from scratch. Probably the most interesting result is the comparison with a hierarchical behaviour cloning network and the experiments around skills transfer. The idea seems sound, although the presentation is in need of some improvement, and at times the paper suffers from lack of clarity (in problem formulation and presentation of results). Moreover, some important work in the admittedly vast array of work on hierarchical models for RL and behaviour cloning is missed.



**Summary Of The Review:**

I think this work has some very interesting experiments and results, particularly around the question of how we should trade-off leveraging existing skills against learning new ones, but these are currently not highlighted very prominently, due to a lack of clarity in problem formulation and experimental results.

There is a wealth of research around hierarchical mixture latent variable policies and skill discovery which is missing from this work, and reduces the strength of the architecture as a contribution. Refocusing on the fact that the architecture makes it easier to develop methods and strategies for transferring to new settings would help to distinguish this paper from this related work.

====== Post rebuttal comments ======

Thank you for your response and for the hard work in addressing the feedback provided. I feel the paper is much stronger for this, so have updated my score.

---

> ### Author Response · Authors · 2021-11-19
> **Authors' Response to Reviewer NRgW**
>
> Thank you for your invaluable feedback; we have made several changes to the paper, including improving the clarity of the method and experiments, and significantly extending the related work to better position our contribution. We address your concerns in detail below.
>
> **The paper is in some need of smoothing, and I found this a difficult read, despite being familiar with the field. In particular, I believe the paper would benefit from a clearer problem formulation (ie. is the problem setting a two stage learning from demonstration one followed by an RL phase that makes use of the initial learned skills, or is it a tabular rasa setting with skills gradually incorporated and re-used) Initial sections suggest the former, but some experimental settings and baselines seem to indicate both. I understand that the architecture is very general, and the aim was to show it can be used in both settings with tweaks, but I think the paper would benefit greatly from making the different settings more clear in an expanded problem formulation.**
>
> Thanks for the feedback. The setting that we focus on is a two-stage process, where the hierarchical skill space is learned from an offline dataset (Section 2.2 of the paper), and reloading and transferring the skills via reinforcement learning (Section 2.3). While the qualitative analyses look at the skills that are learned offline, the quantitative experiments focus on the transfer performance, since the ultimate goal is to learn skills that can transfer effectively to downstream tasks. Most of the baselines follow this two-stage setting, with the exception of MPO, which represents the setting of learning online from-scratch. We regret that this was not clear, and have expanded the problem formulation and modified the experimental discussion to rectify this.
>
> **I would also greatly appreciate a more structured experiments setting, more clearly delineating the tasks and with a much more focused aim. As an example, the baselines are all designed to solve slightly different problems, and the broad set of experiments are not necessarily presented to make a cohesive argument, rather they seem to aim to answer many separate questions. A potential option would be to restructure this section around the question of how much prior do we need and how quickly we can allow skill adaptation, which is a very interesting research question.**
>
> While these questions (how much prior we need and how fast we can allow skill adaptation) are relevant and interesting avenues for future work, they are somewhat orthogonal to our key questions that we state at the start of section 4. We aim to demonstrate what we argue is a key utility of learned skills: that a single skill space can be flexibly and effectively be reused for different tasks and realistic transfer scenarios that may arise, improving sample efficiency and performance across many of these cases. Thus, the key questions are centred around whether we can (i) learn a hierarchical skill space combining discrete and continuous modes of behaviour, (ii) use it for sample-efficient transfer in different scenarios, and (iii) improve understanding of where such skills are useful; and we structure the experiments to answer these questions in turn.
> We have made changes to hopefully clarify this more cohesive perspective in the experimental section.
>
> **Existing literature in hierarchical and reusable skill discovery for RL/ continuous control**
>
> Thank you for providing this extensive list of existing literature - we have incorporated them into the related work section to better position our work.
>
> **Minor queries and comments**
>
> The number of skills are pre-specified for this paper. We have fixed the paper to clarify this and have addressed the other minor issues.

---

> > ### Comment · Reviewer_NRgW · 2021-11-21
> > **Thanks, is some more discussion on Florensa et al warranted?**
> >
> > Thank you for your response and the significant improvements you have made to the paper, it addresses many of my concerns.
> >
> > However, I still have some concerns about the framing of the papers experiments and a clearer distinction between this and prior work. My comment
> >
> > > A potential option would be to restructure this section around the question of how much prior do we need and how quickly we can allow skill adaptation, which is a very interesting research question.
> >
> > was not intended to be prescriptive, but rather a route to a potential improvement to make the contributions and new findings clearer. A more nuanced statement of contribution and acknowledgement of prior work would help readers to understand why they should read this work, and what they would learn from it.
> >
> > As it stands, I think many of the questions you set out to ask are already answered to a large extent in prior work. Florensa et al. - ICLR 17 https://openreview.net/forum?id=B1oK8aoxe is an excellent example of this.
> >
> > Specifically:
> > > Thus, the key questions are centred around whether we can (i) learn a hierarchical skill space combining discrete and continuous modes of behaviour,
> >
> > Florensa et al. investigate architectures that explore the integration of both discrete and continuous latent variables in a neural network for hierarchical skill learning, answering this question. Architecturally, the differences I see here are the inclusion of dynamics on the discrete latent variable and the addition of a skip connection between input and state in the policy.
> >
> > > (ii) use it for sample-efficient transfer in different scenarios, and (iii) improve understanding of where such skills are useful; and we structure the experiments to answer these questions in turn. We have made changes to hopefully clarify this more cohesive perspective in the experimental section.
> >
> > Florensa et al's experiments investigates transfer from an improved exploration perspective, alongside where these are useful. While I appreciate that your analysis is more detailed and covers a much better set of tasks, it is still worth acknowledging that many of these questions have already been answered in prior work in this section, and that your work is adding to this existing evidence.
> >
> > By my reading of prior work, we already know that hierachical architectures work, that they can discover useful skills and improve exploration and sample efficient learning, in sparse reward settings. We also know that they can slow down learning in other settings, depending on the environments. What we don't know is what architectures are good for this, and how we should train or reuse these skills to get the best out of our models, and balance the tradeoff between using prior skills and learning new ones. For me, these are the key strengths and contributions of your work.

---

> > > ### Author Response · Authors · 2021-11-22
> > > **We have made various changes to the paper to better position our work**
> > >
> > > Thank you for taking the time to provide this detailed response. We have carefully gone through and re-read Florensa et al (2017) in the context of your comments. We have added details to the related work section to better place and distinguish our contributions, changed the wording of our experimental questions to be more specific to the proposed hierarchical architecture, and added context to the exploration/sparse reward analysis to note that it adds to previously observed evidence. We hope the revisions have adequately addressed your concerns on this, but are always grateful to receive constructive feedback.

---

### Official Review · Reviewer_wAVF · 2021-11-03

**Correctness:** 4
**Technical Novelty And Significance:** 3
**Empirical Novelty And Significance:** 2
**Recommendation:** 8
**Confidence:** 4

**Main Review:**

Strengths:
- The paper is well-written and easy to follow. It is nice to read (except for some parts of the method that are rushed)
- The method is novel, extending prior work
- The results show some improvements

Weaknesses:
- Structurally, I like papers that push the related work section to the back because they first present some relevant common theoretical components to understand both the presented method and the related work. This is not the case here. I would recommend placing the related work after the introduction to fully understand and compare the method section to the previous methods.
- The experimental evaluation is scarce. The method, although explained as very general, is only applied to one domain. Some of the results are not completely clear. The paper would gain on clarity and support for conclusions if the method would be applied to other RL domains, even simple ones. Trained policies could be used to generate expert demonstrations. In its current form, the experiments fall short.
- The comparison to other methods, especially to NPMP, is not completely clear. The results are somewhat mixed. This is probably a consequence of the limited experimental evaluation with only one domain. Right now, apart from the sparse reward setup, it is not very clear the pros and cons of the presented method compared to previous ones.
- Could you provide an intuition of what is the different information represented by the categorical and continuous high-level latent codes? Both represent the context. What is exactly the benefit of the two levels? Temporal consistency and commitment? Semantic information encoding? Additional experiments in other domains and with other mixtures of information at different levels would help
- What happens if there are other observations passed to the different levels? How sensitive are the results to that?
- What happens if during training one or more of the categorical values are not allowed? Can the system recover (find the necessary skills via exploration)?
- I’d recommend including a small figure with the objects used in the experimental evaluation to avoid having to go back and forth to know what are “set1”, “set2”...
- The method section is a bit rushed. I’d dedicate more time to go step by step over the derivation (not completely, that is in the appendix), explaining more of why things are done instead of just describing the mathematical equations.


**Summary Of The Paper:**

This paper presents a method to learn a three-leveled hierarchy of skills offline, from a dataset of demonstrations, that can then be applied to accelerate reinforcement learning. The three-level architecture is novel. It encodes a discrete selection, a continuous contextual variable (dependent on the discrete selection) and a low level policy dependant of the continuous contextual variable. The results show some improvements over baselines, especially in a sparse reward context where the presented method capitalizes from the offline learned strategies for exploration.

**Summary Of The Review:**

In summary, the paper presents a novel approach with three hierarchical levels for skill learning. The approach is novel, the results are good, the text is well-written, and the problem is relevant. The experimental evaluation is rather limited and several questions arise with respect to the benefits over previous approaches, the robustness and generality.

---

> ### Author Response · Authors · 2021-11-19
> **Authors' Response to Reviewer wAVF (1/2)**
>
> Thank you for the insightful comments; we have made several changes to the paper, including improving the clarity of the method and experiments, and adding further analysis and discussion to the appendix. We address your concerns in detail below.
>
> **Structurally, I like papers that push the related work section to the back because they first present some relevant common theoretical components to understand both the presented method and the related work. This is not the case here. I would recommend placing the related work after the introduction to fully understand and compare the method section to the previous methods.**
>
> Our related work section does actually appear after the method section, and the cited works in the introduction are mostly to provide context and scaffolding for the motivation for this work (eg. to motivate the use of both continuous and discrete skill representations). Could you clarify your comments on this?
>
> **The experimental evaluation is scarce. The method, although explained as very general, is only applied to one domain. Some of the results are not completely clear. The paper would gain on clarity and support for conclusions if the method would be applied to other RL domains, even simple ones. Trained policies could be used to generate expert demonstrations. In its current form, the experiments fall short.**
>
> We planned to run experiments with other offline datasets and environments (such as D4RL or RL-unplugged) to extend the manipulation experiments. Unfortunately these won’t be ready before the end of the discussion period, but we hope to complete and include this analysis for the camera-ready submission.
>
> **The comparison to other methods, especially to NPMP, is not completely clear. The results are somewhat mixed. This is probably a consequence of the limited experimental evaluation with only one domain. Right now, apart from the sparse reward setup, it is not very clear the pros and cons of the presented method compared to previous ones.**
>
> From our experiments, the key takeaways related to other approaches are that it can significantly improve sample efficiency and asymptotic performance for different transfer scenarios (ie. tasks, objects, state-to-vision), and that it does so by performing more directed exploration via the mid-level skills, which is especially beneficial in sparse reward tasks (as shown by our additional analysis). Our ablation study is also intended to show the benefit of both discrete and continuous latent variables in the approach, which also shows the pros when compared to NPMP (continuous-only). We have now clarified these key points in the paper, and as mentioned above, we hope to finish experiments on a second domain before the camera-ready version. Are there any other specific aspects for which you felt the comparison was unclear?
>
> **Could you provide an intuition of what is the different information represented by the categorical and continuous high-level latent codes? Both represent the context. What is exactly the benefit of the two levels? Temporal consistency and commitment? Semantic information encoding? Additional experiments in other domains and with other mixtures of information at different levels would help**
>
> The key intuition is that we can capture coarse discrete modes of behaviour with the categorical, and vary the execution of each of these behaviours with the continuous latent variable. While we don’t enforce temporal consistency in skill learning or reuse, we find that having the learned categorical transition prior uncovers temporal consistency in behaviours; actively encouraging and exploiting temporal consistency is an interesting direction for future work. We have changed the text in the paper to make all of this more clear.
>
> **What happens if there are other observations passed to the different levels? How sensitive are the results to that?**
>
> In our previous experiments we found that without information-asymmetry, such that the mid-level is conditioned on lookahead information as well, the mixture can degenerate to a single or few components and transfer poorly. This is consistent with previous work that shows that information-asymmetry with task-specific information is necessary to encode information into a high-level controller.
> We will re-run these ablations to ensure a fair comparison with the latest model, and include the results in the appendix of the paper for the camera-ready version.

---

> > ### Author Response · Authors · 2021-11-19
> > **Authors' Response to Reviewer wAVF (2/2)**
> >
> > **What happens if during training one or more of the categorical values are not allowed? Can the system recover (find the necessary skills via exploration)?**
> >
> > Intuitively, this would depend on the capacity of the model during offline learning, in terms of the number of categorical high-level skills. In earlier experiments, we found that with greater numbers of skills, multiple components would be used to represent similar behaviours, which would be more robust to components being removed during transfer. This is something that could be better explored in future work.
> >
> > **[Changes to method section]**
> >
> > We have adapted the method section to include a figure of the objects, and spend more time explaining the model equations and decisions, with additional detailed discussion in the appendix.

---

> > > ### Comment · Reviewer_wAVF · 2021-11-22
> > > **Response to rebuttal**
> > >
> > > Thank you. I think the manuscript improved in clarity and scope. I adapted my score

---

> > ### Comment · Reviewer_wAVF · 2021-11-19
> > **Quick reply on the structure**
> >
> > "Our related work section does actually appear after the method section...Could you clarify your comments on this?"
> >
> > Right now, you explain your method, then related work. However, a significant part of your method is "background", things that prior work has done. The current form of the text makes it hard to understand what is prior and what is novel from this work. One option: Background -> RW -> Method. Or RW -> Method (highlighting the new parts). Or Background+RW together -> Method.
> >
> > Or simply RW -> Method. Then the reader already reads the method understanding the similarities and differences.

---

> > > ### Author Response · Authors · 2021-11-22
> > > **We have reorganised the paper to address your comment**
> > >
> > > Thank you for your prompt clarification; this helps us to better understand your original suggestions. We have now reorganised the paper such that the related work appears before the method, and added some more pointers to highlight the novel aspects of HeLMS. We hope this makes the contribution more clear to readers.

---

### Author Response · Authors · 2021-11-19
**Authors' response to all reviewers, and summary of changes**

We are grateful to the reviewers for their insightful and constructive feedback, which we have taken on board to significantly improve the paper. From our understanding, reviewers were generally positive about the idea, but felt that there could be more clarity and discussion around the method and experiments, and that the work could be better positioned with respect to prior art. As a result, we have made numerous changes (highlighted in blue in the revised submission), including:
- Improving the equations and technical depth, with more explanation and justification (including additional details about both offline and online training in the appendix).
- Additional clarity around the method and setting in this paper.
- Extending the related work to capture the extensive literature in skill segmentation and movement primitives.
- Additional discussion of experimental observations, and further analysis in the appendix.

We hope we have satisfactorily addressed the reviewers’ concerns, and would be happy to receive any further suggestions.

---

> ### Author Response · Authors · 2021-11-22
> **Authors' summary of changes #2**
>
> The reviewers were quick to respond with further clarifications and comments, and we thank them for closely engaging with our paper. We have made a second round of edits to address their comments and suggestions, including fixing typos, reorganising the paper to discuss related work early, and making our contributions w.r.t prior work more clear throughout.

---

### Public Comment · ~Haonan_Yu5 · 2022-01-29
**Questions regarding online RL**

Hi, nice work and congratulations!
While reading the paper, I saw in the appendix that the "trajectory length" is 10 for online RL (Table 5). Is this number just the episode length (time limit) for the online RL tasks? (If yes, then it seems kind of short, given the complex manipulation task?). Another related question is, it seems to me that for the three-level hierarchy, currently there is no temporal abstraction yet. Namely, at every time step, all levels will output actions, although their actions have different semantic abstractions (within a time step). Is my understanding correct? Thanks.

---

> ### Public Comment · ~Dushyant_Rao1 · 2022-03-16
> **Answers to your questions**
>
> Thanks for the kind words, and apologies for the long delay in replying - I somehow missed your comment!
>
> To answer your questions:
> - The trajectory length is not the duration of the episode, but the length of trajectories stored by the actor and used by the learner. ie. for each learner update, the data batch is of size n_batch x traj_length x (state/action_dim).
> - That's correct, there isn't any explicit temporal abstraction upon reuse. Given that the learned skills seem to exhibit temporal consistency (as shown by visualisations and the learned prior), this may be useful as well.
>
> Hope this helps; please let us know if you have further questions or comments!

---

### Decision · Program_Chairs · 2022-01-20

**Decision:**

Accept (Spotlight)

**Comment:**

This work propose to learn hierarchical skill representations that, as opposed to prior work, consist of both discrete and continuous latent variables. Specifically, this work proposes to learn 3 level hierarchy via a hierarchical mixture latent variable model from offline data. For test time usage and adaptation on down-stream tasks, the manuscript proposes two ways of utilizing the learned hierarchy in RL settings.

**Strengths**
A novel method to learn hierarchical representations with mixed (discrete/continuous) latent variables is proposed
Detailed experimental evaluation, and baseline comparisons, show promising results

**Weaknesses**
There were various clarity issues as pointed out by the reviewers (fixed in rebuttal phase)
The related work was missing relevant work, and the proposed framework was not connected well to existing work (fixed during rebuttal)

**Rebuttal**
The authors significantly updated the manuscript based on the feedback of the reviewers, and improved both clarity of the manuscript (method+experiments) as well as the exposition of the proposed framework with respect to related work.

**Summary**
After the rebuttal, all reviewers agree that this is a good paper that should be accepted. Thus my recommendation is accept.